# MSX1 Regulates Goat Endometrial Function by Altering the Plasma Membrane Transformation of Endometrial Epithelium Cells during Early Pregnancy

**DOI:** 10.3390/ijms24044121

**Published:** 2023-02-18

**Authors:** Beibei Zhang, Zongjie Wang, Kangkang Gao, Rao Fu, Huatao Chen, Pengfei Lin, Aihua Wang, Yaping Jin

**Affiliations:** 1Key Laboratory of Animal Biotechnology of the Ministry of Agriculture, College of Veterinary Medicine, Northwest A&F University, Yangling 712100, China; 2Department of Clinical Veterinary Medicine, College of Veterinary Medicine Northwest A&F University, Yangling 712100, China; 3Department of Preventive Veterinary Medicine, College of Veterinary Medicine, Northwest A&F University, Yangling 712100, China

**Keywords:** MSX1, CDH2, plasma membrane transformation, goat endometrial epithelial cell, endoplasmic reticulum stress

## Abstract

MSX1 is an important member of the muscle segment homeobox gene (Msh) family and acts as a transcription factor to regulate tissue plasticity, yet its role in goat endometrium remodeling remains elusive. In this study, an immunohistochemical analysis showed that MSX1 was mainly expressed in the luminal and glandular epithelium of goat uterus, and the MSX1 expression was upregulated in pregnancy at days 15 and 18 compared with pregnancy at day 5. In order to explore its function, goat endometrial epithelial cells (gEECs) were treated with 17 β-estrogen (E_2_), progesterone (P_4_), and/or interferon-tau (IFNτ), which were used to mimic the physiological environment of early pregnancy. The results showed that MSX1 was significantly upregulated with E_2_- and P_4_-alone treatment, or their combined treatment, and IFNτ further enhanced its expression. The spheroid attachment and PGE2/PGF2α ratio were downregulated by the suppression of MSX1. The combination of E_2_, P_4_, and IFNτ treatment induced the plasma membrane transformation (PMT) of gEECs, which mainly showed the upregulation of N-cadherin (CDH2) and concomitant downregulation of the polarity-related genes (*ZO-1*, *α-PKC*, *Par3*, *Lgl2*, and *SCRIB*). The knockdown of MSX1 partly hindered the PMT induced by E_2_, P_4_, and IFNτ treatment, while the upregulation of CDH2 and the downregulation of the partly polarity-related genes were significantly enhanced when MSX1 was overexpressed. Moreover, MSX1 regulated the CDH2 expression by activating the endoplasmic reticulum (ER) stress-mediated unfolded protein response (UPR) pathway. Collectively, these results suggest that MSX1 was involved in the PMT of the gEECs through the ER stress-mediated UPR pathway, which affects endometrial adhesion and secretion function.

## 1. Introduction

The receptive endometrium and filamentous conceptus are essential for implantation [1,2]. Two-thirds of failed implantations are due to the fact of abnormal endometrial receptivity [3]. During endometrial receptivity formation, the endometrial epithelium undergoes ultrastructural and biochemical alterations of the plasma membrane, which is vital for embryo survival and implantation [4,5,6]. The plasma membrane transformation (PMT) is characterized by changes in the surface adhesion proteins and the loss of luminal epithelial cell polarity, which is a common phenomenon across species [7,8,9,10]. However, the mechanisms that regulate the PMT are not well understood, especially in ruminants.

The polarity of endometrial epithelial cells (EECs) is regulated by hormone [11] and maintained by polarity complexes containing apical (atypical PKC, Stardust, and Crumbs), basolateral determinants (Scribble), tight junctions, and adhesion junctions, which directly affect cell adhesion and secretory function [12,13]. The relevant research shows that the PMT of EECs confers adhesive and secretory capacity in humans [7], mice [14,15], and rats [6,16]. In the human secretory phase of the menstrual cycle, the polarity markers are downregulated by ovarian hormones in the luminal and glandular epithelium layer [8]. Moreover, several transcription factors and signaling molecules, such as MSX1, KLF5, Stat3, and FGFR2, have been implicated in playing important roles in the PMT of EECs, which can be affected by ovarian hormones [14,17,18]. In ruminants, some genes involved in endometrial remodeling are also stimulated by ovarian hormones during the peri-implantation period and further enhanced by IFNτ secreted from conceptus [19,20], yet the mechanisms underlying how hormones and conceptus signals regulate PMT remain elusive.

Homeobox transcription factor MSX1, a DNA-binding protein, plays a critical role in embryogenesis and skeletal development, but its functions are limited in most adult tissues due to the reduction of plasticity [21]. However, the female endometrium is an exception for which it can undergo cellular, morphological, and molecular changes during endometrial receptivity. *MSX1* deletion of uterus in female mice resulted in the reduction of the embryo implantation rate due to the disturbance of endometrial dynamics [18]. The reduction of MSX1 in human endometrial tissue is linked to infertility [22]. Furthermore, BMP2 acts on MSX1 to regulate endometrial decidualization in mice and humans [23]. In ovine uterus, dynamic endometrial changes are primarily regulated by maternally derived 17 β-estrogen (E_2_) and progesterone (P_4_), and it is enhanced by conceptus-derived interferon-tau (IFNτ) [24].

During embryo implantation, the endometrium undergoes differentiation from prereceptivity to receptivity in ruminants [25]. The development of endometrial receptivity is known as the “window of implantation” from the attachment of conceptus to the completion of adhesion [26,27]. On day 5 of pregnancy in goats, the endometrium was in a prereceptivity stage. The endometrium was in a receptivity stage on day 15 [2,28]. The conceptus adhered firmly to the receptivity endometrium on day 18 of pregnancy in goats [29]. However, the localization and expression of MSX1 has not been reported in the uterus of goat, especially in the peri-implantation period. Moreover, whether MSX1 regulates the dynamic changes of gEECs remains poorly understood. Additionally, it is still unclear whether MSX1 can be induced by E_2_, P_4_, and IFNτ to participate in the dynamic changes of gEECs.

## 2. Results

### 2.1. Immunolocalization and Expression of MSX1 in the Endometrial Tissue of Goats

Representative images of MSX1 immunolocalization in goat endometrial tissue at pregnancy at days 5 (P5, n = 3), 15 (P15, n = 3), and 18 (P18, n = 3) are shown in Figure 1A. The immunohistochemical staining of MSX1 in endometrial tissue showed that MSX1 was mainly localized in LE (Figure 1d–f), sGE (Figure 1d–f), and GE (Figure 1g–i), while a low expression level was detected in the stromal cells (S) (Figure 1d–f). Compared with P5, the immunoreactivity of MSX1 was upregulated at P15 and P18, the mRNA expression of MSX1 was significantly increased at P15 and P18 (Figure 1B, *p* < 0.01), and the protein expression of MSX1 was also significantly increased at P15 and P18 (Figure 1C, *p* < 0.05). 

### 2.2. MSX1 Was Upregulated with Hormone and/or IFNτ Treatment in the gEECs

To further explore the biological function of MSX1, the gEECs were treated with E_2_, P_4_, and/or IFNτ to mimic the intrauterine environment during the peri-implantation period. The expression of MSX1 in the gEECs was detected with E_2_, P_4_, and/or IFNτ treatments. The results showed that the expression of MSX1 was upregulated in the E_2_, P_4_, E_2_ + P_4_, and E_2_ + P_4_ + IFNτ groups compared with the control group, and the upregulation in the E_2_ + P_4_ + IFNτ group was more significant than the other groups (Figure 2A,B, *p* < 0.01). Furthermore, MSX1 was localized abundantly in the cytoplasm and nucleus of the gEECs under E_2_, P_4_, and IFNτ treatment compared to the control group, and the fluorescence was upregulated (Figure 2C, *p* < 0.05). These results indicate that MSX1 was induced in the gEECs by E_2_ and P_4_ and was further enhanced with IFNτ treatment. 

### 2.3. Knockdown of MSX1 Affected Endometrial Adhesion and Secretory Function

The mRNA and protein levels of MSX1 in the shMSX1 group were significantly lower than that in the shN group (Appendix A, *p* < 0.01). *ISG15*, *RSAD2*, and *CXCL10* were detected for the acting marker genes of endometrial receptivity in goats, the mRNA level of these genes decreased in the shMSX1 group compared with the shN group (Figure 3A, *p* < 0.01), and the attachment of the GTC spheroids in the shMSX1 group was also lower than the shN group (Figure 3B, *p* < 0.01). To further confirm the results, the key cell adhesion molecule of SPP1 was measured, and similar results were obtained (Figure 3C,D). The concentration of PGF_2α_ in the shMSX1 group was higher than the shN group (Figure 3F, *p* < 0.05). However, the secretion of PGE_2_ remained unchanged (Figure 3E, *p* > 0.05). Further, the mRNA levels of *PTGS2* and *PGFS* were upregulated by inhibiting MSX1 (Figure 3H, *p* < 0.01), but *PTGS1* and *PTGES* had no statistically significant difference. Compared with the shN group, the PGE_2_/PGF_2α_ ratio was lower in the shMSX1 group (Figure 3G, *p* < 0.05). These results show that the knockdown of *MSX1* affected the endometrial adhesion and secretion function, further disturbing the uterus receptivity formation.

### 2.4. Hormones and/or IFNτ Induced the PMT of gEECs 

The PMT of endometrial epithelial cells was induced by ovarian hormones, and further enhanced by IFNτ, which altered the adhesion and secretory of the epithelium to prepare for conceptus implantation. In the present study, the mRNA and protein expression of CDH2 was significantly increased in the E_2_ + P_4_ and E_2_ + P_4_ + IFNτ groups compared with control group, and this trend was more pronounced in the E_2_ + P_4_ + IFNτ group (Figure 4A,B, *p* < 0.01). The immunofluorescence results show a significant difference between the two conditions, and the fluorescence intensity of the cytoplasmic CDH2 in the gEECs was enhanced in the E_2_ + P_4_ + IFNτ group (Figure 4C). Furthermore, the mRNA expression of the polarity markers (*ZO-1*, *α-PKC*, *Par3*, *Lgl2*, and *SCRIB*) decreased (Figure 4D, *p* < 0.05). 

### 2.5. Knockdown of MSX1 Hindered the PMT with Hormones and IFNτ Treatment

The knockdown of MSX1 decreased the expression of CDH2 with E_2_, P_4_, and IFNτ treatment in the gEECs (Figure 5A,B, *p* < 0.01), and the mRNA expression of the polarity markers (*ZO-1*, *α-PKC*, *Par3*, *Lgl2*, and *SCRIB*) was upregulated to various degrees (Figure 5C, *p* < 0.05).

### 2.6. Overexpression of MSX1 Enhanced the PMT 

Compared with the empty vector group, the mRNA and protein levels of CDH2 were upregulated when MSX1 was overexpressed in the gEEC cells (Figure 6A,B, *p* < 0.01), and the cell polarity (*α-PKC*, *Par3*, *Lgl2*, and *SCRIB*) was decreased in the MSX1 overexpression group (Figure 6C, *p* < 0.05). However, the mRNA expression level of the tight junction protein *ZO-1* did not significantly change (Figure 6C, *p* > 0.05). These results indicate that MSX1 may affect endometrial adhesion and secretion function through regulated PMT.

### 2.7. MSX1 Regulated the Expression of CDH2 via the ER Stress-Mediated UPR Pathway

As the ER stress-mediated UPR pathway plays an important role in the regulation of endometrial function, the UPR marker proteins were examined, and the results show that the expressions of phosphoEIF2S1, phosphoIRE1, and XBP1 were downregulated in the shMSX1 group compared with the shN group (Figure 7A, C, *p* < 0.05). However, cleaved ATF6 expression showed limited change (Figure 7B, *p* > 0.05). Moreover, the inhibitor (4-PBA) and activator (Tg) were used to alter the ER stress, and it was found that the expression of GRP78 significantly decreased in the 4-PBA group, while it was significantly upregulated in the Tg group (Figure 7D, *p* < 0.01). The CDH2 protein expression was downregulated after inhibiting MSX1 expression. The downregulation was significantly enhanced in the shMSX1 group pretreated with 4-PBA. In contrast, the activation of ER stress significantly compensated for the decrease in the CDH2 expression when *MSX1* was knocked down (Figure 7D, *p* < 0.05). These results suggest that MSX1 regulated the expression of CDH2 by modulating the ER stress-mediated UPR pathway.

## 3. Discussion

The mechanisms of endometrium receptivity formation are particularly important to determine the causes of goat embryo mortality. Uterine receptivity refers to the physiological state of the uterus during embryonic development and implantation during pregnancy at 18 days in goat [26]. Compared with goat uterus during pregnancy at 5 days, the MSX1 expression was upregulated gradually in pregnancy at 15 and 18 days, which suggests that MSX1 may be involved in the formation of endometrium receptivity. Uterine receptivity is primarily regulated by ovarian hormones, and it is further enhanced by conceptus signals in ruminants [30]. Combinations of E_2_, P_4_, and IFNτ were used to mimic the hormonal environment during early pregnancy in goat [31]. In the present study, we found that the expression of MSX1 in the gEECs was significantly upregulated with E_2_, P_4_, and IFNτ treatment, which showed that MSX1 was expressed in a dynamic temporal and depended on E_2_ and P_4_ regulation, and it was enhanced by IFNτ.

In ruminants, some genes that can promote conceptus elongation are considered markers for endometrial receptivity formation [25,32], such as *ISG15*, *CXCL10,* and *RSAD2,* and can be induced by hormones of maternal secretion and stimulated by conceptus-derived IFNτ. In this study, these marker genes were downregulated when *MSX1* was silenced, which showed that MSX1 was indispensable for the endometrial receptivity formation of goat. The endometrial LE is the first contact site of embryonic trophoblasts, and its secretions and adhesion functions are essential for conceptus survival and implantation [33]. The spheroid coculture assay was widely applied to measure the cell adhesion in vitro [15,34]. The percentage of attached GTC spheroids in the gEECs decreased meaningfully with the downregulation of MSX1. In ruminants, polarized EECs secrete prostaglandins (PGs) during early pregnancy, which are essential for embryo implantation [35,36,37]. Endometrial PGF_2α_ is responsible for the luteolysis, while PGE_2_ is thought to be involved in the maternal recognition of pregnancy (MRP) [38]. The function of the two PGs is opposite during MRP, so the ratio between PGE_2_ and PGF_2α_ is important for endometrial receptivity [30,31]. Notably, the secretion of PGF_2α_ was enhanced, and the PGE_2_/PGF_2α_ ratio was decreased by silencing *MSX1*. These results suggest that MSX1 positively regulated endometrial adhesion and secretion functions, and it further affected endometrial receptivity formation.

The PMT describes the collective morphological and molecular alterations that occur in the endometrial LE for the facilitation of the implantation [7], which confers the adhesive and secretion capacity of the LE, and this is essential for embryo implantation. Some studies suggest that PMT has multiple parallels with epithelial–mesenchymal transition (EMT) [4,7], such as the upregulation of N-cadherin (CDH2) expression. CDH2 is both a member of the cadherin superfamily and a mesenchymal cell maker, and its upregulation is as a hallmark of EMT [39]. Uchida’s research found that the adhesion of JAR (a human choriocarcinoma cell line) spheroids can induce a cadherin switch in Ishikawa cells with E_2_ and P_4_ treatment [15]. In ruminants, some genes were involved in the endometrial remodeling of embryo implantation, which are stimulated by ovarian hormones and further enhanced by IFNτ during the maternal recognition period [40]. In the present study, the expression of CDH2 was upregulated in the E_2_ + P_4_ group and E_2_ + P_4_ + IFNτ group. Moreover, polarity-related genes, including *ZO-1*, *α-PKC*, *Par3*, *Lgl2*, and *SCRIB*, showed a significant increase in the E_2_ + P_4_ + IFNτ group. These results indicate that the maternal hormonal environment may induce PMT, which was further enhanced by conceptus-derived IFNτ. 

The knockout of Msx1 in mice uteri reduced the embryo implantation rate due to the disturbance of the endometrial dynamics [18]. In addition, the decrease in the MSX1 expression in human endometrial tissue is linked to infertility [22]. In this study, the expression pattern of MSX1 was the same as CDH2, with E_2_-, P_4_-, and IFNτ-alone treatment and their combination. Meanwhile, the adhesion and secretion capacity of the gEECs changed when MSX1 was silenced. Interestingly, MSX1 overexpression enhanced the expression of CDH2, and the mRNA levels of polarity-related genes were partially downregulated, except for *ZO-1*. In the gut tissues of humans and mice, estrogen directly regulates the tight junction protein ZO-1 by activating the membrane receptor (GPER) [41,42]. Therefore, we suspect that E_2_ may regulate the expression of ZO-1 in gEECs, which may be parallel to the MSX1 signaling pathway. These results indicate that MSX1 affected endometrial function by regulating the PMT of gEECs induced by hormones and enhanced by conceptus signals.

The endometrium is a highly plastic tissue that undergoes dynamic remodeling, and the ER plays a pivotal role in the synthesis, folding, and modification of cell surface protein [43]. However, the accumulation of unfolded or misfolded proteins in the ER lumen leads to stress, which causes the activation of the UPR [44]. Many studies have shown that ER stress-mediated UPR signaling cascades play an important role in mammalian gestation [45]. Moreover, E_2_, P_4_, and IFNτ were used to mimic the intrauterine environment during the peri-implantation period of goats, which can induce ER stress and activate the UPR [34]. Previous studies have shown that phosphoEIF2S1, phosphoIRE1, cleaved ATF6, and XBP1 can be used to investigate the potential mechanism of ER stress-mediated physiological phenomena [46]. In the present study, the expression of phosphoEIF2S1, phosphoIRE1, and XBP1 was downregulated in the shMSX1 group. Tg (ER stress agonist) and 4-PBA (ER stress inhibitor) were used to change the state of the ER. The GRP78 protein acts a monitor for ER stress [47]. In the present study, the decrease in CDH2 was compensated with the 50 nM Tg treatment when the MSX1 gene was silenced in the gEECs, while it was opposite with the 1 mM 4-PBA treatment. Taken together, these results suggest that the MSX1 regulation of the remodeling of the gEECs via ER stress-UPR signaling is plausable.

## 4. Materials and Methods

### 4.1. Tissue Collection

Multiparous Guanzhong dairy goats (n = 9, age = 2–3 years, and average weight = 59.28 ± 1.93 kg) were sampled at the Experimental Animal Center of Northwest A&F University, Yangling, China. The goats exhibiting at least two estrous cycles of normal duration were used in this study. At estrus, female goats were mated with fertile males to induce natural pregnancy, which was recorded as day 0 of pregnancy. Pregnancy was confirmed on day 5 by recovering blastocysts from the uterus. Pregnancy at day 15 and day 18 was, respectively, identified by observing the elongated tubular conceptus and fibrous conceptus in the uterus. The uterine tissues were collected from receptively, pregnancy at day 5 (P5, n = 3, pre-receptivity stage), pregnancy at day 15 (P15, n = 3, receptive stage), and pregnancy at day 18 (P18, n = 3, conceptus adhesion period) and immediately fixed in 4% paraformaldehyde. All experimental procedures were performed in accordance with the Committee for the Ethics on Animal Care and Experiments of Northwest A&F University (Approval No. 2019100903).

### 4.2. Immunohistochemistry

Paraffin sections containing uterine tissue were deparaffinized and hydrated in graded ethanol series before staining with the streptavidin–peroxidase method. The antigens were retrieved by boiling for 15 min in citrate antigen retrieval solution (Solarbio, Beijing, China). Endogenous peroxidase was blocked by incubation in 3% hydrogen peroxide. Then, the instructions of the universal SP staining kit were followed (Maixin Biotechnologies, Fuzhou, China). Briefly, the sections were pretreated with 0.3% H_2_O_2_ for 40 min at 37 °C to quench the endogenous peroxidase activity; then, they were washed with phosphate-buffered saline (PBS). The sections were incubated with 10% serum for 60 min at 37 °C. After blocking, the sections were incubated with anti-MSX1 antibody (1:100, Abcam, ab168745, Cambridge, UK) overnight at 4 °C, and the second antibody was incubated at 37 °C for 2 h. The sections were then washed with PBS, then incubated with streptavidin-biotin peroxidase for 40 min at 25 °C. Thereafter, the sections were visualized with diaminobenzidine (DAB), lightly counterstained with hematoxylin for 25 s, dehydrated, and coverslipped. As a negative control, the primary antibody was substituted with pre-immune serum. The sections were imaged under a microscope (Nikon, Tokyo, Japan) after drying at 25 °C. 

### 4.3. Cell Culture and Treatment

Goat endometrial epithelial cells (gEECs) and goat trophoblast cells (gGTCs) were immortalized by transfection with human telomerase reverse transcriptase [48,49]. The gEECs were positively stained for the epithelium marker keratin, and the gEECs showed normal goat chromosome numbers after passage number 50 [50]. The gEECs were cultured in DMEM/F-12 medium containing 10% FBS (Corning, Manassas, VA, USA) and incubated at 37 °C in a humidified 5% CO_2_ incubator. When the gEECs reached a 50–60% confluence, the schedule of the gEECs is show in Appendix A. In short, the medium was replaced with fresh DMEM/F-12 and 0.1% bovine serum albumin (BSA; R&D Systems, Inc., Minneapolis, MN, USA) was added for 24 h. The gEECs were then treated with P_4_ (10^−7^ M; Sigma, St. Louis, MO, USA) and/or E_2_ (10^−9^ M, Sigma, St. Louis, MO, USA) for 12 h. IFNτ (20 ng/mL, Sangon Biotech Co., Ltd., Shanghai, China) was added to the medium for 6 or 12 h [34]. In the drug group, 50 nM thapsigargin (Tg, ER stress activator; Sigma, St. Louis, MO, USA) and 1 mM 4 phenyl butyric acid (4-PBA, ER stress inhibitor; Sigma, St. Louis, MO, USA) were added to the gEECs before adding IFNτ for 1 h.

### 4.4. In Vitro Implantation Assay

In vitro embryo implantation was simulated using a spheroid coculture assay, performed according to adapted instructions from previous studies [34,51]. The concentration of GTCs was adjusted to 2 × 10^5^ cells/mL to generate GTC spheroids as blastocyst models. The cell suspension was stained with CellTracher Green CMFDA (0.5 μM; Yeasen Biotech Co., Ltd., Shanghai, China) and cultured on an orbital shaker rotating at 200 rpm for 10 h at 37 °C. Spheroids approximately 100–200 mm in diameter, which were similar in size to those of an implanting blastocyst, were obtained through a 100-mesh sterilized sieve. The trophoblastic spheroids were prepared and gently delivered onto monolayer gEECs, which were cultured in 24-well plates and treated with E_2_, P_4_, and IFNτ for 6 h, as per the cell culture described above. After the EECs reached confluence, gEECs and GTC spheroids (approximately 100 spheroids per well) were cocultured with fresh medium for 1 h at 37 °C. The 24-well culture plates were washed with PBS three times. The GTC spheroids were counted under a fluorescence microscope. The attachment rate is expressed as the percentage of seeded spheroids. 

### 4.5. RNA Extraction and Real-Time Quantitative PCR

The total RNA was extracted from the gEECs using TRIzol reagent (TaKaRa, Tokyo, Japan), according to the manufacturer’s instructions. The RNA concentration and purity were measured using an Ultramicro spectrophotometer (ThermoScientifc, Waltham, MA, USA), following the PrimerScript^TM^RT reagent kit (TaKaRa, Tokyo, Japan) instructions, and the RNA was converted into complementary DNA (cDNA). Total RNA = 0.4 μg, 5 ×PrimeScript RT Master Mix = 10 μL, and RNase Free ddH20 = up to 20 μL were used. The primer sequences are shown in Appendix A. The amplification curve and melt curve of primers are shown in Appendix A. Real-time quantitative PCR (qPCR) was performed using SYBR Green Master Mix (Vazyme, Nanjing, China) in a Bio-Rad CFX96 (Bio-Rad, Hercules, CA, USA), according to the manufacturer’s protocol. cDNA = 2 μL, 2 ×PrimeScript RT Master Mix = 10 μL, forward primer and reverse Primer (10 μM) = 0.8 μL, RNase Free ddH20 = up to 20 μL were used. The RT-PCR parameters were as follows: 37 °C for 15 min and 85 °C for 5 s. The qPCR parameters were as follows: 95 °C for 30 s, followed by 40 cycles each of 95 °C for 5 s and 60 °C for 20 s. The level of mRNA quantification was estimated with the 2^−∆∆ct^ method. The expression of mRNA was normalized to the *GAPDH* gene, which served as the control gene in all samples.

### 4.6. Western Blot 

The samples were lysed with RIPA buffer (Solarbio, Beijing, China), according to the manufacturer’s protocol. The total protein concentration was measured using the BCA assay (Keygen Biotech, Nanjing, China). The total protein (20 μg) was loaded onto a 12% SDS-PAGE gel. The proteins were then transferred to PVDF membranes (Millipore; Bedford, MA, USA). After blocking with 10 % nonfat milk for 2 h, the samples were incubated with primary antibodies at 4 °C overnight; they were as follows: anti-MSX1 antibody (1:1000, ab168745, Abcam, Cambridge, UK), anti-CDH2 antibody (1:1000, ab76057, Abcam, Cambridge, UK), anti-EIF2S1 (phosphorS51) antibody (1:1000, ab32157, Abcam, Cambridge, UK), anti-EIF2S1 antibody (1:1000, ab26197, Abcam, Cambridge, UK), anti-IRE1 (phosphorS724) antibody (1:1000, ab124945, Abcam, Cambridge, UK), anti-XBP1 antibody (1:1000, ab37152, Abcam, Cambridge, UK), anti-GRP78 antibody (1:1000, 3183, CST, Boston, MA, USA), anti-ATF6 antibody (1:1000, ab83504, Abcam, Cambridge, UK), and anti-β-actin antibody (1:5000, Sanjian Biotech, Wuhan, China). Subsequently, the membranes were incubated for 1 h with an HRP-labeled secondary antibody at room temperature. Finally, the protein bands were visualized using a gel imaging system (1:5000, Tannon Biotech, Shanghai, China) and quantified using Quantity One software (Bio-Rad Laboratories, Hercules, CA, USA).

### 4.7. Immunofluorescent Staining

The samples were from cell sections and fixed with 4% formaldehyde for 30 min, rinsed three times with PBS, and permeabilized with PBST (0.1% Triton X-100 in PBS) for 5 min. After blocking with 3% BSA for 1 h and washing three times, the samples were incubated overnight at 4 °C; anti-MSX1 antibody (1:100, ab168745, Abcam, Cambridge, UK), anti-CDH2 antibody (1:100, ab76057, Abcam, Cambridge, UK), and anti-SPP1 antibody (1:150, WL02378, Wanlei Biotech, Shenyang, China) were used. After washing three times with PBS, incubation with Alexa-labeled donkey anti-mouse IgG (1:500, A16016, Invitrogen, Waltham MA, USA,) and Alexa-labeled donkey anti-rabbit IgG (1:500, A16028, Invitrogen, Waltham MA, USA) was performed at room temperature for 1 h. The nuclei were counterstained with 4’,6-diamidino-2-phenylindole for 5 min, and the cells were observed using laser-scanning confocal microscopy (Nikon, Melville, NY, USA). The negative control refers to incubating only with the Alexa-labeled secondary antibody, which are shown in Appendix A.

### 4.8. Cell Transfection with the Interference Target Sequence and MSX1 Plasmid

The goat MSX1 gene (GeneID: 102190677) cDNA was amplified and subcloned into the GV362 (Genechem Science and Technology Ltd., Shanghai, China) plasmid, resulting in the construction of the GV362-MSX1 plasmid. The gEECs were transiently transfected with an empty vector or GV362-MSX1 using Turbofect Transfection Reagent (Thermo Scientific, Waltham, MA, USA), according to the manufacturer’s instructions. 

We constructed short hairpin RNA lentiviral vectors for the targeted MSX1 gene, named shMSX1, and a pair of oligonucleotides expressing a scrambled sequence, named shN. The target sequences are listed in Appendix A. The virus packaging and cell transfection were performed as described previously [52]. 

### 4.9. Prostaglandin Measurement

The gEECs infected with shMSX1 or shN were counted using a cell count plate, plated in 24-well plates (5 × 10^4^ cells/well), and cultured as described above. After incubation with IFNτ for 12 h, the culture supernatants were collected. The concentration of prostaglandinF_2α_ (PGF_2α_) and prostaglandin E_2_ (PGE_2_) were measured by the Goat Prostaglandin F_2α_ ELISA kit (JYM, Wuhan, China) or Goat Prostaglandin E_2_ ELISA kit (JYM), according to the manufacturer’s instructions.

### 4.10. Statistical Analysis 

Unless otherwise specified, all data are expressed as the mean ± S.E.M., with no less than three replicates for each experimental condition. One-way analysis of variance followed by Tukey’s post hoc test and Fisher’s LSD test were used for multiple comparisons. Statistical differences were considered significant when the *p*-value was less than 0.05. 

## 5. Conclusions

In summary, MSX1 was mainly expressed in the luminal and glandular epithelium of goat uterus, and MSX1 expression was gradually enhanced during early pregnancy. The combination of E_2_, P_4_, and IFNτ treatment induced the upregulation of MSX1 and PMT in the gEECs. MSX1 was involved in the regulation of PMT in the gEECs through the ER stress-mediated UPR pathway, ultimately affecting endometrial adhesion and secretion functions.

## Figures and Tables

**Figure 1 ijms-24-04121-f001:**
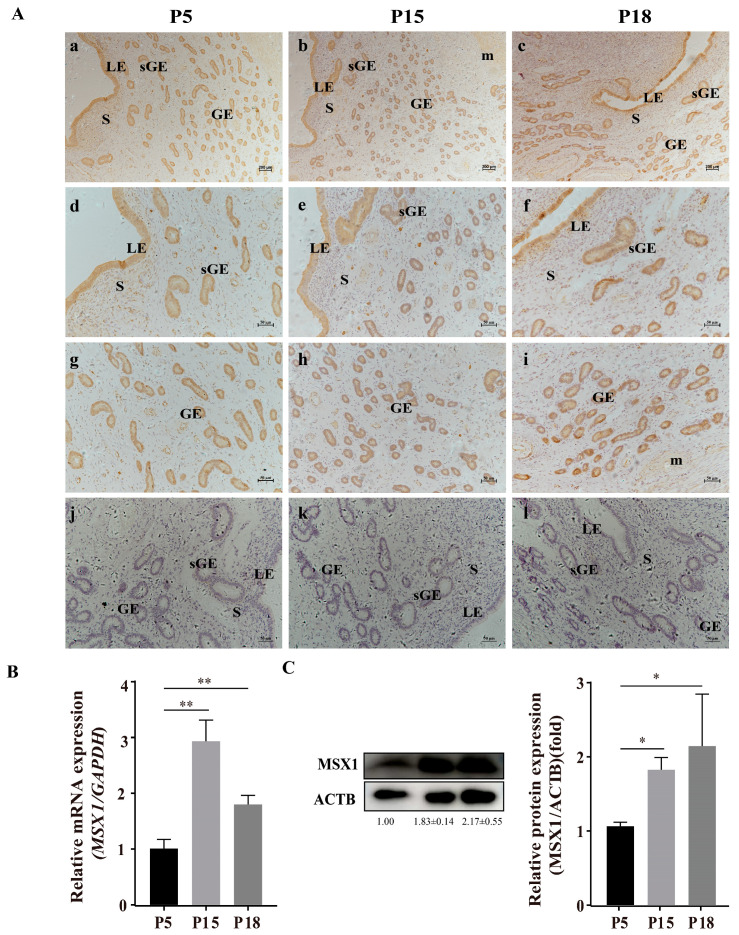
Representative images of MSX1 immunolocalization and expression in goat endometrial tissue at pregnancy at days 5, 15, and 18. (**A**) Representative images of MSX1 immunolocalization in goat endometrial tissue at pregnancy at days 5, 15, and 18: (**a**,**d**,**g**) MSX1 immunolocalization in goat endometrial tissue on P5 (n = 3); (**b**,**e**,**h**) MSX1 immunolocalization in goat endometrial tissue on P15 (n = 3); (**c**,**f**,**i**) MSX1 immunolocalization in goat endometrial tissue on P18 (n = 3); (**j**–**l**) negative control in goat endometrial tissue on P5, P15, and P18. (**B**) The mRNA expression level of MSX1 in P5, P15, and P18. (**C**) Western blot and analysis of MSX1 protein in P5, P15, and P18. LEs: luminal epithelial cells; sGE: superficial glandular epithelium; GEs: glandular epithelial cells; S: stromal cells; m: myometrium. (**a**–**c**) scale bar = 200 μm; (**d**–**l**) scale bar = 50 μm. * *p* < 0.05; ** *p* < 0.01.

**Figure 2 ijms-24-04121-f002:**
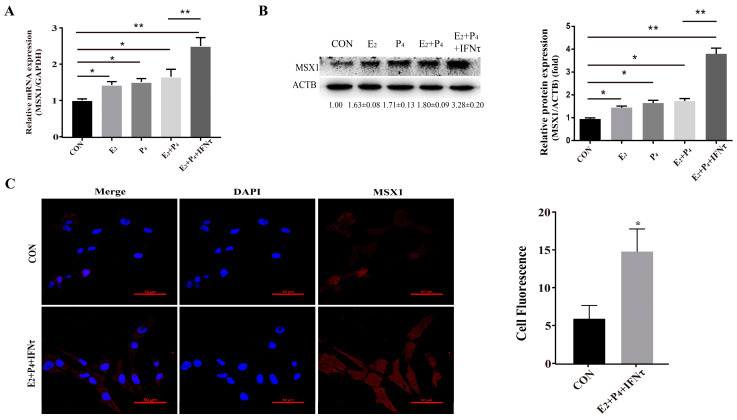
Hormones and/or IFNτ treatment induced MSX1 expression. (**A**) mRNA expression level of *MSX1* with E_2_, P_4_, and/or IFNτ treatment; (**B**) Western blot and analysis of MSX1 protein levels with E_2_, P_4_, and/or IFNτ treatment; (**C**) representative images and analysis of the MSX1 location of gEECs in the Con and E_2_ + P_4_ + IFNτ groups. Scale bar = 50 μm. * *p* < 0.05; ** *p* < 0.01.

**Figure 3 ijms-24-04121-f003:**
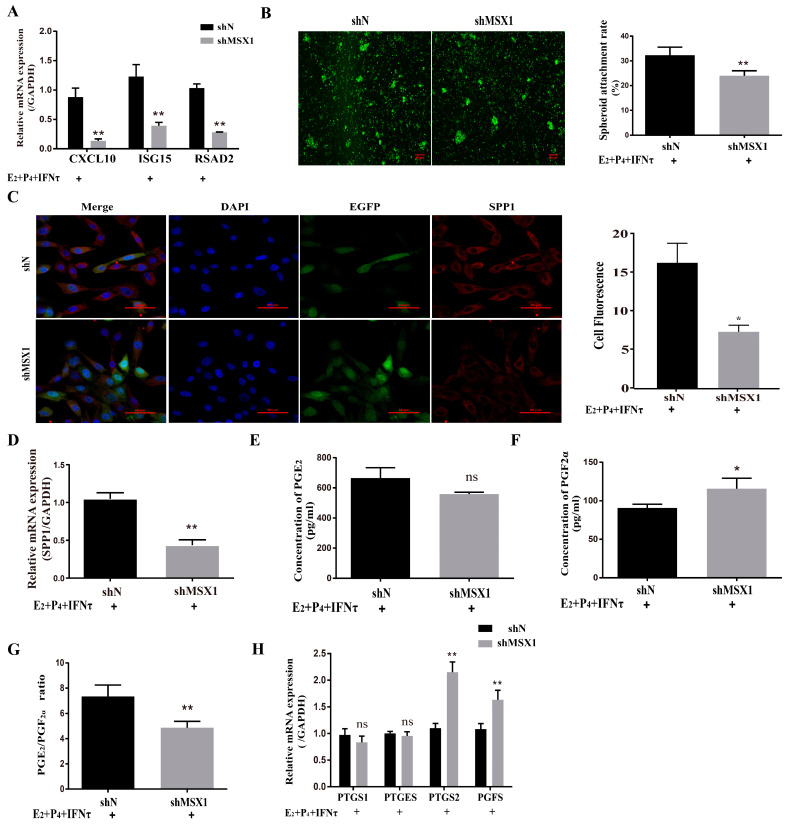
Knockdown of MSX1 affected endometrial adhesion and secretory function. (**A**) mRNA expression level of uterine receptivity markers (*ISG15, CXCL10,* and *RSAD2*) in the shN group and shMSX1 group; (**B**) representative images and analysis of GTC spheroids in the shN group and shMSX1 group, scale bar = 50 μm; (**C**) representative images and analysis of SPP1 expression in the shN group and shMsx1 group, scale bar = 50 μm; (**D**) mRNA expression level of *SPP1* in the shN group and shMSX1 group; (**E**,**F**) secretion of PGE_2_ and PGF_2α_ in the shN group and shMSX1 group; (**G**) ratio of PGE_2_/PGF_2α_ in the shN group and shMSX1 group; (**H**) mRNA expression level of the PGs synthesis genes in the shN group and shMSX1 group. ns: *p* > 0.05; * *p* < 0.05; ** *p* < 0.01.

**Figure 4 ijms-24-04121-f004:**
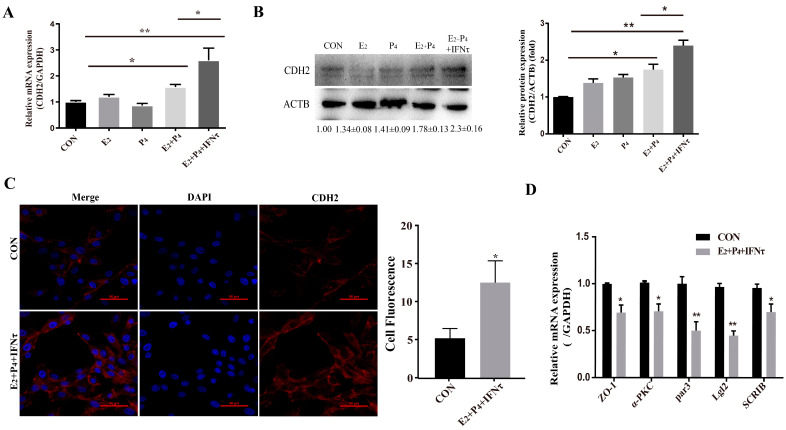
Hormones and/or IFNτ induced the PMT of gEECs. (**A**) mRNA level of *CDH2* with E_2_, P_4_, or/and IFNτ treatment; (**B**) Western blot and quantitative analysis results of CDH2 with E_2_, P_4_, or/and IFNτ treatment; (**C**) representative images and analysis of CDH2 expression, scale bar = 50 μm; (**D**) mRNA levels of polarity-related genes: *ZO-1*, *α-PKC*, *Par3*, *Lgl2*, and *SCRIB*. * *p* < 0.05; ** *p* < 0.01.

**Figure 5 ijms-24-04121-f005:**
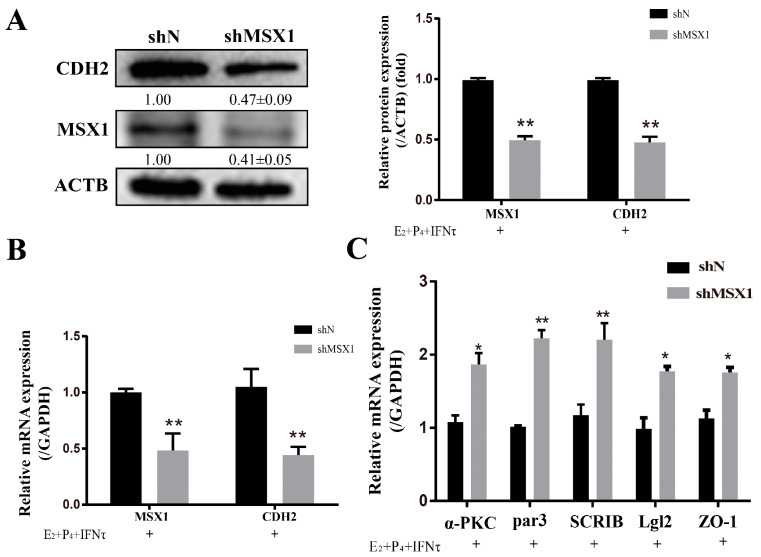
Knockdown of MSX1 hindered the PMT induced by hormones. (**A**) Western blot and analysis results of CDH1 and MSX1 expression in the shN group and shMSX1 group; (**B**) mRNA expression levels of *CDH2* and *MSX1*; (**C**) mRNA expression levels of *ZO-1*, *α-PKC*, *Par3*, *SCRIB,* and *Lgl2*. * *p* < 0.05; ** *p* < 0.01.

**Figure 6 ijms-24-04121-f006:**
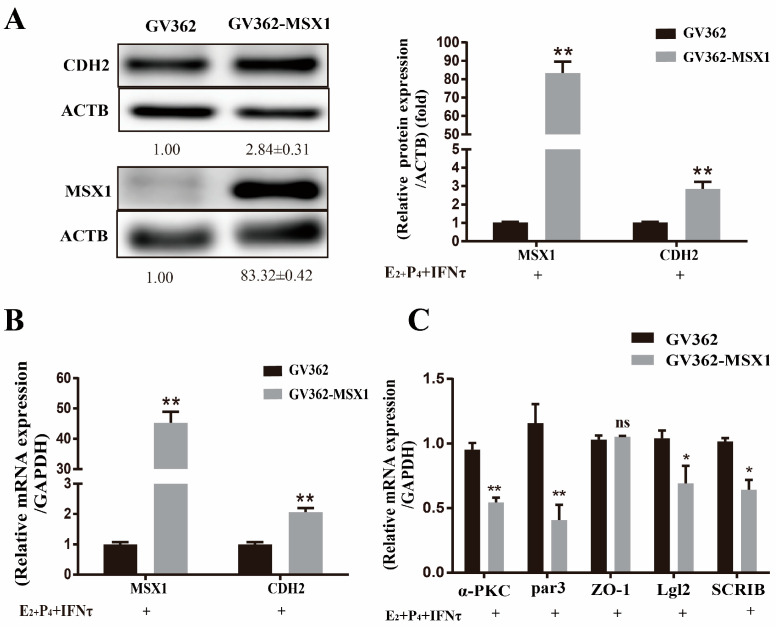
Overexpression of MSX1 enhanced the PMT. (**A**) Western blot and analysis results of CDH1 and MSX1 expression in the GV362-MSX1 group and GV362 plasmid group; (**B**) mRNA expression levels of *CDH2* and *MSX1* in the GV362-MSX1 group and GV362 plasmid group; (**C**) mRNA expression levels of *ZO-1*, *α-PKC*, *Par3*, *SCRIB,* and *Lgl2* in the GV362-MSX1 group and GV362 plasmid group. ns: *p* > 0.05; * *p* < 0.05; ** *p* < 0.01.

**Figure 7 ijms-24-04121-f007:**
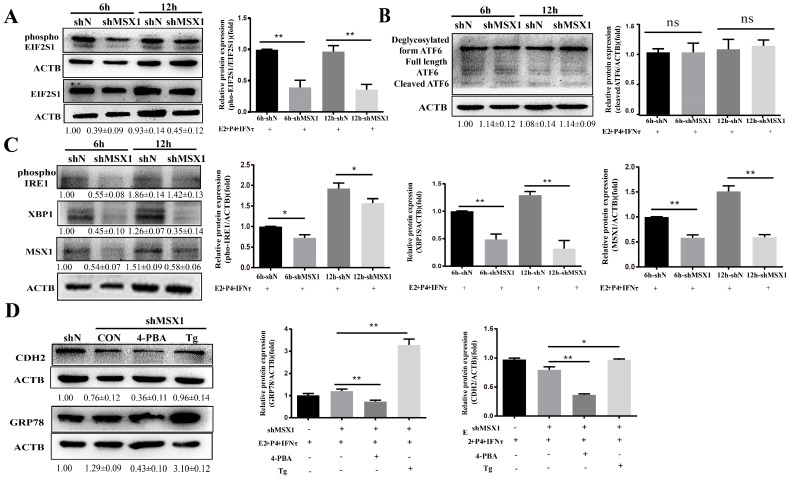
MSX1 regulated the expression of CDH2 via the ER stress-mediated UPR pathway. (**A**) Western blot and analysis results of EIF2S1and phosphoEIF2S1; (**B**) Western blot and analysis results of the cleaved and deglycosylated forms of ATF6; (**C**) Western blot and analysis results of phosphoIRE1 and XBP1; (**D**) shMSX1 EECs were treated with P_4_ and E_2_, and they were pretreated with TG or 4-PBA followed by treatment with IFNτ for 6 h, showing the Western blot and quantitative analysis results of GRP78 and CDH2. ns: *p* > 0.05; * *p* < 0.05; ** *p* < 0.01.

## Data Availability

Not applicable.

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
