# Peer review of "MSX1 Regulates Goat Endometrial Function by Altering the Plasma Membrane Transformation of Endometrial Epithelium Cells during Early Pregnancy"

_ijms, 2023, doi:10.3390/ijms24044121_

Round 1
Reviewer 1 Report
- In this study, the role of MSX1 in goat endometrium remodeling was investigated at the cellular and animal levels, which has important implications for the field.
But there are still some issues to be solved.
-
1. Figure 1: WB and mRNA levels of MSX1 should be increased.
- 2. Figure 2. Adding gray value labeling and mRNA level detection results;
- 3.
WB picture quality is poor and needs to be improved.
- 4.
The full-text language needs to be proofread and revised by native English speakers.
Author Response
We would like to thank the reviewer for their accurate and detailed revision of our manuscript. Reviewer offered helpful and constructive suggestions and as a result, we feel that the manuscript has been significantly improved. We have seriously thought about the suggestions and provided our responses in a point-by-point manner.
Comments
In this study, the role of MSX1 in goat endometrium remodeling was investigated at the cellular and animal levels, which has important implications for the field. But there are still some issues to be solved.
Response: Thanks for reviewer’s positive comments and constructive suggestions of the work, we have seriously thought about the suggestions and provided our responses in a point-by-point manner, are given as follows.
Point 1: Figure 1: WB and mRNA levels of MSX1 should be increased.
Response 1: Thanks for reviewer’s kindly reminding and advice. In the early stage of this study, the localization and expression abundance of MSX1 gene in goat endometrium were mainly observed, and it was found that MSX1 was mainly expressed in the luminal epithelium and glandular epithelium. The abundance of MSX1 gene expression in the uterus of goats at 15 and 18 days gestation was higher than that in the uterine cavity and glandular epithelium of goats at 5 days gestation. Endometrium is not only composed of luminal epithelium and glandular epithelium, but also composed of stroma, we did not further detect the gene and protein expression levels of MSX1 at 5, 15 and 18 days of gestation for that we do not have good technology to clearly separate luminal epithelium, glandular epithelium and stroma. In order to explore the function of MSX1 in the early stage of embryo implantation, we used goat endometrial epithelial cells as the research object for further in-depth study.
Point 2: Figure 2. Adding gray value labeling and mRNA level detection results;
Response 2: Thanks for reviewer’s advice, and we have adding gray value labeling in Figure 2B, the mRNA level detection results were show in figure 2A, the suggestions have revised in the manuscript
Point 3: WB picture quality is poor and needs to be improved.
Response 3: Thanks for reviewer’s kindly reminding and advice. There is no doubt that WB picture quality is really poor and needs to be improved. In the manuscript, all the WB picture was visualized using the Gel Image System(Tannon Biotech, Shanghai, China), yet the output WB picture quality is almost the same, so we sincerely apologize for this, but we will try our best to improve in the manuscript, and other pictures were improved, such as Fig 1J, Fig 1K, Fig 1L, Fig 2C, Fig 3C, Fig 4C .
Point 4: The full-text language needs to be proofread and revised by native English speakers.
Response 4: Thanks for reviewer’s kindly reminding and advice. We regret there were problems with the English. The manuscript has been polished before submission(the prove has been uploaded), but the manuscript may didn’t been carefully revised, so we have revised it seriously again as suggested, and carefully checked it.
The main revisions are as follow:
Abstract
1.Deleting“In ruminant, endometrium undergoes dynamic remodeling during pre-implantation, which is essential for the elongation and implantation of conceptus. ”.
2.Adding “mmunohistochemical analysis showed that MSX1 was mainly expressed in the luminal and glandular epithelium of goat uterus, MSX1 expression was upregulated in pregnancy day 15 and 18 compared with pregnancy day 5” ..
3.Deleting “MSX1 expression was upregulated during early pregnancy of goat”.
4.Deleting “under normal physiological conditions”.
5.Adding “17 β-estrogen, progesterone, interferon-tau, which were used to ”.
6.Deleting “hormonal, in”.
7.Adding “physiological, of”
8.Adding “alone treatment ”.
9.Deleting “than the combination”
10.Deleting “which disturbed the uterus receptivity formation. Receptivity is a polarity-dependent function of hormone-regulated uterine epithelial cells.”
11.Adding “treatment, of” .
- “par3” has revised into “Par3”
13.Deleted “by, hormone” .
14.Adding “E2, P4, treatment”.
15.Deleting “the expression of”.
16.Adding “expression”
17.Adding “ER, UPR”
List of abbreviastions
1.Adding “Msh: muscle segment homeobox gene; E2: 17 β-estrogen; P4: progesterone; IFNτ: interferon-tau; gEECs: goat endometrial epithelial cells; PMT: plasma membrane transformation; CDH2: N-cadherin; ER: endoplasmic reticulum; UPR: unfolded protein response; gGTCs: goat trophoblast cells; LE: luminal; sGE: superficial glandular epithelium” in the list of abbreviastions
Introduction
1.“In ruminant, the” was revised into “The”
2.A space was added before citing the reference in the whole manuscript
3.Deleting “is the first contact site for embryonic trophoblasts to”
4.“undergo” was revised into “undergo”
5.“in” was revised into “of”
6.Adding “and”
7.Adding “which is”
8.Deleting “and shows strikingly”
9.“Phenomena” was revised into “phenomenon”
10.Deleting “precise”, “in different species”
11.Deleting “the endometrial receptivity is a”, “dependent function of hormone-regulated”
12.Adding “The”, “of endometrial”, “(EECs is regulated by hormone and)”
13.Deleting “The polarity of endometrial epithelial cell(EEC) is”
14.Deleting “and”
15.Deleting “The uterine epithelia are present in the endometrium of all mammals”
16.“endometrial epithelial cells” was revised into “EECs”
17.Adding “of menstrual cycle”
18.Adding “by ovarian hormones in ”
19.Deleting “within”
20.Deleting “under the influence of ovarian hormones”
21.Deleting “induced by hormones”
22.“endometrial epithelial cells” was revised into “EECs”
23.Adding “that”, “also”
24.“However” was revised into “yet”
25.Deleting “in ruminants”
26.“the” was revised into “its”
27.“with reduced” was revised into “due to the reduction”
28.“yet” was revised into “However”
29.“that” was revised into “is an exception for which it can undergo”
30.Deleting “is one exception”
31.“of ” was revised into “in”
32.“reduced” was revised into “resulted in the reduction”
33.Deleting “before implantation”, “homeobox protein”
34.Adding “BMP2 acted”, “to”
35.Deleting “homeobox gene”
36.Deleting “via downstream signaling of BMP2”
37.Deleting “However, whether MSX1 regulates the dynamic changes in EECs of goat remains poorly understood.’
38.“ruminants” was revised into “ovine uterus”
39.Adding “17 β-estrogen” and deleting “estradiol”
40.“which” was revised into “and”
41.Adding “During embryo implantation, the endometrium undergoes differentiation from pre-receptivity to receptivity in ruminant. The development of endometrial receptivity is known as the “window of implantation” from the attachment of conceptus to the completion of adhesion. On day 5 of pregnancy in goats, the endometrium was in a pre-receptivity stage. The endometrium was in a receptivity stage on day 15. The conceptus adheres firmly to the receptivity endometrium on day 18 of pregnancy in goats. However, the localization and expression of MSX1 has not been reported in uterus of goat, especially in P5, P15, P18. Moreover, whether MSX1 regulates the dynamic changes of gEECs remains poorly understood.However, the localization and expression of MSX1 has not been reported in uterus of goat, especially in P5, P15, P18. Moreover, whether MSX1 regulates the dynamic changes of gEECs remains poorly understood. Additionally, whether MSX1 can be induced by E2, P4 and IFNτ to participate in dynamic changes of gEECs remain unclear. ”
42.Deleting “mimic the vivo intrauterine environment during peri-implantation period to explore the role of MSX1 in endometrial receptivity formation of goat. These results indicated that MSX1 was involving in regulating plasma membrane transformation of endometrial epithelial cells induced by hormones and INFτ, and affected goat endometrial function during early pregnant, which provided new insights about the PMT of EECs during endometrial receptivity in goat. ”
Materials and methods
2.1 Tissue Collection
1.Adding “Multiparous Guanzhong”
2.Adding “(n=9, aged 2-3 years, average weight=59.28±1.93kg,)”
3.Adding “The goats exhibiting at least two estrous cycles of normal duration were used in this study. At estrus, female goats were mated with fertile males to induce natural pregnancy, which was recorded as day 0 of pregnancy. Pregnancy was confirmed on day 5 by recovering blastocysts from the uterus. Pregnancy at day 15 and day 18 was respectively identified by observing the elongated tubular conceptus and fibrous conceptus in the uterus. ”
4.“uterus” was revised into “uterine”
5.Deleting “pregnancy”
6.Adding “pregnancy”
7.“day 5 (P5, n=3), day 15 (P15, n=3) and day 18 (P18, n=3) of pregnancy” was revised into “pregnancy day 5 (P5, n=3, pre-receptivity endometrium), pregnancy day 15 (P15, n=3, receptive endometrium) and pregnancy day 18 (P18, n=3, conceptus adhesion period)”
8.Adding “Approval No. 2019100903”
2.2 Immunohistochemistry
- Adding “(Maixin Biotechnologies, China). Briefly, the sections were pretreated with 0.3% H2O2 for 40 min at 37℃ to quench endogenous peroxidase activity, then washing with phosphate-buffered saline (PBS), the sections were incubated with 10% serum for 60 min at 37℃. After blocking, the sections were incubated with,”
- Deleting “in which the primary antibody”
- “in which the primary antibody anti MSX1 (1:100, Abcam USA) is incubated ” was revised into “anti-MSX1 antibody (1:100, Abcam, ab168745, USA)”
- Adding “The sections were then washed with PBS, then incubated with streptavidin-biotin peroxidase for 40 min at 25°C. Thereafter, the sections were visualized with diaminobenzidine (DAB), lightly counterstained with hematoxylin for 25 s, dehydrated, and coverslipped. As a negative control, the primary antibody was substituted with pre-immune serum. Sections were imaged under a microscope (Nikon, Germany) after drying at 25°C. ”
- Deleting “After incubation, sections were lightly counterstained with hematoxylin and were dehydrated and cover slipped. ”
2.3 Cell culture and treatment
- “EECs” was revised into “gEECs”
- “GTCs” was revised into “gGTCs”
- Adding “for 6h or 12h”
2.4 In vitro implantation assay
- “study” was revised into “researches”
- Deleting “gently”, “phosphate buffered saline”
- Deleting “remaining in the wells”
2.5 RNA extraction and real-time quantitative PCR
- “PrimerScripyt” was revised into “PrimerScript”
- Adding “Total RNA 0.4 μg, 5 ×PrimeScript RT Master Mix 10 μL, RNase Free ddH20 up to 20 μL.”
- “Real-time quantitative PCR(RT-qPCR)” was revised into “Real-time quantitative PCR(qPCR)”
- Adding “cDNA 2 μL, 2 ×PrimeScript RT Master Mix 10 μL, Forward Primer and Reverse Primer(10μM) 0.8 μL, RNase Free ddH20 up to 20 μL. The RT-PCR parameters were as follows: 37 °C for 15 min, 85 °C for 5 s. The qPCR parameters were as follows: 95 °C for 30 s, followed by 40 cycles each of 95 °C for 5 s, 60 °C for 20 s.”
2.6 Western blot
- Adding “20 μg”
- Deleting “and the proteins were allowed to separate byelectrophoresis”
- Deleting “in Tris buffered saline containing 0.5% Tween 100”
- Adding “primary antibodies at 4 ℃ for overnight, they were as follows”
- “anti-MSX1 antibody (Abcam ab168745), anti-CDH2 antibody (Abcam ab76057), anti-EIF2S1(phosphorS51) antibody (Abcam ab32157), anti-IRE1 (phosphorS724) antibody (Abcam ab124945), anti-XBP1 antibody (Abcam ab37152), anti-GRP78 antibody (CST 3183), and anti-β-actin antibody (Tianjin Sanjian Biotech Co., Ltd.) ” were revised into “anti-MSX1 antibody (1:1000, ab168745, Abcam, USA), anti-CDH2 antibody (1:1000, ab76057, Abcam, USA), anti-EIF2S1 (phosphorS51) antibody (1:1000, ab32157, Abcam, USA), anti-EIF2S1 antibody (1:1000, ab26197, Abcam, USA), anti-IRE1 (phosphorS724) antibody (1:1000, ab124945, Abcam, USA), anti-XBP1 antibody (1:1000, ab37152, Abcam, USA), anti-GRP78 antibody (1:1000, 3183, CST, USA), anti-ATF6 antibody (1:1000, ab83504, Abcam, USA), anti-β-actin antibody (1:5000, Sanjian Biotech, Wuhan, China)”
2.7 Immunofluorescent staining
- Adding “from cell sections and”
- Deleting “for 5 min”
- Deleting “by incubation”
- “in” was revised into “with”
- Adding “overnight at 4 ℃”
- Adding “anti-MSX1 antibody (Abcam, ab168745, USA, 1:100), anti-CDH2 antibody (1:100, ab76057, Abcam, USA,), anti-SPP1 antibody (1:150, WL02378, Wanleibio, Co., Ltd,)”
- Adding “and alexa-labeled donkey anti-rabbit IgG ( 1:500, A16028, Invitrogen, Waltham MA, USA) ”
2.8 Cell transfection with interference target sequence and MSX1 plasmid
- Adding “named shMSX1 (recombinant lentiviral vectors encoding the MSX1 shRNA) ”
- Adding “named shN (negative control short hairpin RNA)
2.10 Statistical analysis
- Adding “with no less than three replicates”
- Deleting “of three samples”
3 Results
3.1 Immunolocalization of MSX1 in the endometrial tissue of goats
- “MSX1 was upregulated in early pregnancy of goat” was revised into “Immunolocalization of MSX1 in the endometrial tissue of goats”
- Adding “Representative images of MSX1 immunolocalization in goat endometrial tissue at pregnancy 5 (P5, n=3), 15(P15, n=3) and 18 (18, n=3) were shown in figure 1A-I.”
- Deleting “To explore the role of MSX1 in formation of uterus receptive during early pregnancy of goat, we investigated the expression and localization of MSX1 in the endometrium on day 5(P5), day 15(P15) and day 18(P18) of pregnancy”
- Adding “of MSX1 in endometrial tissue”
- “in luminal (LE), superficial glandular epithelium (sGE) and glandular epithelial cells (GE), while a small amount of expression was detected in the stromal cells(S) (Fig 1A). The protein level of MSX1 was significantly upregulated at P18 compared with P15 and P5.” was revised into “in LE(Fig. 1D-F), sGE(Fig. 1D-F) and GE(Fig. 1G-I), while a small amount of expression was detected in the stromal cells (S) (Fig. 1D-F). Compared with P5, the immunoreactivity of MSX1 was upregulated at P15 and P18.”
3.2 MSX1 was upregulated with hormone and/or INFτ treatment in gEECs
- “MSX1 was upregulated with hormone and or INFτ treatment in gEECs” was revised into “MSX1 was upregulated with hormone and/or INFτ treatment in gEECs”
- Deleting “To further explore the biological function of MSX1, the gEECs were treated with E2, P4, and/or INFτ to mimic the intrauterine environment during the peri-implantation period[28, 31].”
- “In the study, we measured ” was revised into “The”
- Adding “was detected with”
3.3 Knockdown of MSX1 affected endometrial adhesion and secretory function
- Deleting “To functionally characterize MSX1 dynamic expression in early gestation of goats, we constructed shN and shMSX1 vectors. As shown in Figure S1A-D,”
- Adding “ISG15, RSAD2, and CXCL10 were detected for acting ”
- Deleting “During early pregnancy, the dynamic changes of endometrial function make the uterus to enter the receptive stage.”
- Deleting “such as ISG15, RSAD2, and CXCL10, which are associated with promoting conceptus elongation. These results showed that”
- “CXCL10, RSAD2, and ISG15” was replaced with “these genes”
- Deleting “The endometrial adhesion function was investigated, and results indicated that”
- Deleting “the expression of SPP1”
- Deleting “by immunofluorescence and real-time quantitative”
- Deleting “were obtained”, “ELISA results demonstrated that”, “in concentration cell supernatant”
- Adding “in shMSX1 group”
- Deleting “the dominant rate-limiting enzymes of synthesize PGs were measured in mRNA level. The results showed that”
3.4 Hormones and/or IFNτ induced the PMT of gEECs
- “During pre-implantation, the PMT of endometrial epithelial cells induced by ovarian hormones, and enhanced by IFNτ alters the adhesion and secretory of the epithelium to prepare for conceptus implantation[4, 18].” was revised into “The PMT of gEECs plays a vital role during pre-implantation. In the present study”
- Deleting “ in which E2 + P4 + IFNτ treatment reduced ”
- Adding “was reduced in E2 + P4 + IFNτ group ”
- Adding “the mRNA expression of polarity markers (ZO-1, α-PKC, Par3, Lgl2, and SCRIB)”
- Deleting “the expression of polarity-related genes were measured under P4, E2, and IFNτ treatments, and the results showed downregulation of tight junction protein ZO-1 and atypical protein kinase C, par3, Lgl2, and SCRIB ”
3.5 Knockdown of MSX1 hindered the PMT with hormones and IFNτ treatment
- “3.4” was revise into “3.5”
- Deleting “To validate whether MSX1 affected endometrial function by regulating PMT, CDH2 expression and the polarity-related genes were detected when MSX1 was knockdown in EECs. These results showed that”
- “following” was revised into “with”
- Adding “in gEECs”
- Adding “the mRNA expression of”
- Deleting “by silencing MSX1”
3.6 Overexpression of MSX1 enhanced the PMT
- “3.5” was revised into “3.6”
- “Compared with the empty vector group”
- Deleting “In contrast to the results of knockdown MSX1”
- Deleting “in compared with the empty vector group”
- Deleting “Genes associated with maintaining”
- “par3” was revised into “Par3’
- Deleting “compared to control groups”
3.7 MSX1 regulated the expression of CDH2 via ER stress-mediated UPR pathway
- “MSX1 regulates the expression of CDH2 via ER stress-mediated UPR pathway ” was revised into “”MSX1 regulated the expression of CDH2 via ER stress-mediated UPR pathway
- “3.6” was revised into “3.7”
- Deleting “in the present study”
- Adding “Moreover”
- Deleting “ In order to investigate whether MSX1 affects PMT of gEECs by regulating ER stress, we altered ER stress using”
- Adding “were used to alter ER stress”
- “groups” was revised into “group”
- “caused by” was revised into “when”
- “knock” was revised into “was knocked”
- Deleting “in part”
- “ER stress” was revised into “ER stress mediated UPR pathway ”
4.Discussion
1.“The characterization of the molecular mechanisms during formation of endometrium receptivity are particularly important to identify the reasons for goat embryo mortality. The endometrium expresses genes associated with uterine receptivity during the first 18 days of pregnancy in goat, which refers to the physiological state of the uterus during embryonic development and implantation” was revised into “The mechanisms of endometrium receptivity formation are particularly important to determine the casue of goat embryo mortality. Uterine receptivity refers to the physiological state of the uterus during embryonic development and implantation during pregnancy 18 days in goat”
2.“In this study, we found that MSX1 protein expression was upregulated gradually on day 5, day15 and day18 of gestation in goat, suggesting that MSX1 may involve in the formation of endometrium receptivity” was revised into “Compared with goat uterus at pregnancy 5 days, MSX1 expression was upregulated gradually in pregnancy 15 and pregnancy 18 days, which suggesting that MSX1 may be involved in the formation of endometrium receptivity.”
3.Deleting “and induced changes of endometrial function”
4.Deleting “In the present study, the combination of E2 and P4 were used to mimic the hormonal environment of the uterus without a conceptus signal, whereas E2, P4, and IFNτ were used to mimic the maternal recognition period. We ”
5.“EECs” was revised into “gEECs”
6.“following” was revised into “with”
7.Deleting “alone or their combination treatment”
8.“Conceptus elongation is a prerequisite for implantation in ruminants. Therefore, some genes promote conceptus elongation induced by hormones of maternal secretion and stimulated by conceptus-derived IFNτ, such as ISG15, CXCL10, and RSAD2, which are considered markers for endometrial receptivity formation” was revised into “In ruminants, some genes that promote conceptus elongation are considered markers for endometrial receptivity formation, such as ISG15, CXCL10, and RSAD2, which can be induced by hormones of maternal secretion and stimulated by conceptus-derived IFNτ. ”
9.“in” was revised into “of”
10.“their” was revised into “its”
11.Deleting “simulated embryo implantation”
12.“to” was revised into “in”
13.Deleting “the regulation of changes occurring in the endometrium prepared for”
14.“recognition of pregnancy” was revised into “recognition of pregnancy(MRP)”
15.“to” was revised into “for”
16.Adding “this”
17.“have” was revised into “has”
18.Adding “both....and”
19.Adding “the expression of CDH2 was upregulated in”
20.Deleting “we found that pre-treatment with both”
21.“as for” was revised into “due to”
22.Deleting “before implantation”
23.Adding “the decrease of” and deleting “reduced homeobox protein”
24.“under” was revised into “with”
25.Deleting “By establishing knockdown (shMSX1) and overexpression (GV362-MSX1) plasmids, inhibition of MSX1 downregulated CDH2 expression and promoted the maintenance of epithelial polarity.”
26.Deleting “response to progesterone and pregnancy”
27.Deleting “a signaling network called the ”
28.Adding “mediated”
29.“In vitro models” was revised into “Moreover”
30.Deleting “ which are ER stress markers”
31.“The results indicate that ER stress induced by 50 nM TG compensated for the decrease in CDH2 expression caused by MSX1 silencing. In contrast, downregulation of CDH2 expression was significantly higher in shMSX1 cells pretreated with 4-PBA. Taken together, these results suggest a plausible role for the MSX1-ER stress-UPR in regulating EEC remodeling. Further studies are required to verify the specific role of the three branches of the UPR in the regulation of epithelial cell remodeling by MSX1, as well as the establishment of more detailed molecular mechanism” was revised into “In the present study, the decrease of CDH2 was compensated with 50 nM TG treatment when MSX1 gene was silenced in gEECs, while it’s opposite with 1 mM 4-PBA treatment. Taken together, these results suggested it’s plausible for the MSX1 regulated the remodeling of gEECs via ER stress-UPR signaling”
- Conclusion
- Adding “MSX1 was mainly expressed in the luminal and glandular epithelium of goat uterus, MSX1 expression was gradually enhanced during ealy pregnancy”
- Deleting “which proved that”
Reference
The style of references were revise.
Figure Legends
1.“Figure 1. MSX1 was upregulated in early pregnancy of goat (A) Immunohistochemical staining of MSX1 protein in goat endometrial tissue on day5(P5), day15(P15) and day18(P18) of pregnancy. LE: luminal epithelial cells, sGE: superficial glandular epithelium, GE: glandular epithelial cells, S: stromal cells, m: myometrium” was revised into “Figure 1. Representative images of MSX1 immunolocalization in goat endometrial tissue at pregnancy 5, 15 and 18. (A,D,G) Representative images of MSX1 immunolocalization in goat endometrial tissue on P5 (n=3)., (B,E,H) Representative images of MSX1 immunolocalization in goat endometrial tissue on P15 (n=3). (C,F,I) Representative images of MSX1 immunolocalization in goat endometrial tissue on P18 (n=3). (G,K,L). Representative images of the negative control in goat endometrial tissue on P5, P15, P18. LE: luminal epithelial cells, sGE: superficial glandular epithelium, GE: glandular epithelial cells, S: stromal cells, m: myometrium; A-C, scale bar=200 μ1m; D-L, scale bar=50 μm.”
2.“Figure 2. Hormones or/and IFNτ treatment induced MSX1 expression” was revised into “Figure 2. Hormones and/or IFNτ treatment induced MSX1 expression. ”
3.“Figure 3. Knockdown of Msx1 affected endometrial adhesion and secretory function. (A) The uterine receptivity markers (ISG15, CXCL10, RSAD2) were measured by real-time quantitative PCR. (B) The GTCs spheroids adhesion were observed by Inverted fluorescence Microscope. Scale bar = 20μm. (C)Confocal microscope images of SPP1 expression in shN and shMsx1groups. (D) The mRNA level of SPP1. (E-F) The secretion of PGE2 and PGF2α. (G) The ratio of PGE2/PGF2α. (I) The relative mRNA expression of synthesize PGs were measured by real-time quantitative PCR. the “ns” as p>0.05; “*” as P<0.05; “**” as P<0.01” was revised into “Figure 3. Knockdown of Msx1 affected endometrial adhesion and secretory function. (A) The mRNA expression level of uterine receptivity markers (ISG15, CXCL10, RSAD2). (B) The representative images of GTC spheroids in shN group and shMSX1 group. Scale bar = 20 μm. (C) The representative images of SPP1 expression in shN group and shMsx1 group. (D) The mRNA expression level of SPP1, (E-F) The secretion of PGE2 and PGF2α. (G) The ratio of PGE2/PGF2α. (I) The mRNA expression level of PGs synthesis genes. the “ns” as p>0.05; “*” as P<0.05; “**” as P<0.01”
4.“Figure 4. Hormones or/ and IFNτ induced the PMT of gEECs. (A) The mRNA level of CDH2. (B)Western blot and quantitative analysis results of CDH2 under E2, P4 or/and IFNτ treatment. (C) Confocal microscope images of CDH2 expression by treatment with E2 and P4 followed with IFNτ 6h. (D) The mRNA levels of polarity related genes ZO-1, α-PKC, par3, Lgl2 and SCRIB. “ns” as p>0.05; “*” as P<0.05; “**” as P<0.01” was revised into “Figure 4. Hormones or/ and IFNτ induced the PMT of gEECs. (A) The mRNA level of CDH2.. (B) Western blot and quantitative analysis results of CDH2 with E2, P4 or/and IFNτ treatment. (C) The representative images of CDH2 expression. (D) The mRNA levels of polarity related genes ZO-1, α-PKC, Par3, Lgl2 and SCRIB. “*” as P<0.05; “**” as P<0.01”
5.In figure 5, “the mRNA level” was revised into “the mRNA expression level” ; “par3” was revised into “Par3” and deleting ‘ “ns” as p>0.05’
6.In figure 6, “the mRNA level” was revised into “the mRNA expresison level” and “par3” was revised into “Par3”
7.In figure 7, “regulates” was revised into “regulated”
Supplementary Materials
1.“Figure S1. Interference efficiency of shMSX1 vectors. (A) The mRNA level of Msx1. (B) The protein level of MSX1 was measured when infected with lentiviral plasmid. (C) Confocal microscope images of MSX1 expression in shN and shMsx1groups. “*” as P<0.05; “**” as P<0.01” was revised into “Figure S1. Interference efficiency of shMSX1 vectors. (A) The representative images and quantitative analysis of MSX1 protein level. (B) The quantitative analysis of MSX1 mRNA level. (C-D) The representative confocal microscopeimages and quantitative analysis of MSX1 expression in shN group and shMsx1group. “*” as P<0.05; “**” as P<0.01”
Reviewer 2 Report
In the present work, Zhang et al. try to explain that MSX1 regulates goat endometrial function by altering plasma membrane transformation of endometrial epithelium cells during early pregnancy. There are some questions that should be explained.
1. Editing of English language and style is needed. Pease revise the manuscript throughout.
For example,
Line 31, ‘epithelial cells(gEECs)’; Line 111, ‘anti-MSX1(1:100,’; Line 214, ‘day5(P5), day15(P15) and day18(P18)’; Line 239, ‘group(p<0.01)’; ……. Please check this throughout the manuscript. A space is needed.
2. Abstract should be rewritten.
Please explain the abbreviations; for example, MSX1, E2, P4, IFNτ and ER….
Lines 29-30, ‘In this study, MSX1 expression was upregulated during early pregnancy of goat.’ This sentence should be rearranged.
Line 40, ‘par3’ should be changed to ‘Par3’.
Line 46, ‘through the ER stress-mediated UPR pathway’. In the Abstract section, ER stress is only present in this sentence.
3. Introduction
Line 71, please explain ‘MSX1, KLF5, Stat3 and FGFR2’.
Line 75, please explain ‘IFNτ’. Interferon-tau, IFNT?
Lines 88-91, In general, the peri-implantation ovine uterus is under the effects of pregnancy, progesterone, and interferon tau (Reference 24). The effect of E2 is before the ovulation, and the level of E2 is low during the peri-implantation. Please explain why treatment with E2.
Lines 92-95, In general, results are not present in Introduction section.
Hypothesis and objective should been added.
4. Materials and methods
Line 99, please add the specie for Dairy goats.
Line 104, please add the number for the Ethics.
Line 110, please add the information for the universal SP staining kit.
Line 111, please add the information for antibody anti-MSX1 (category number), and is this antibody suitable for detecting goat MSX1?
Line 112, please add the DAB staining.
Lines 164-167, are the antibodies suitable for detecting goat proteins?
5. Results
In general, Results section only presents results, but no citing references. Please detect the beganing sentences. For example, detect ‘To explore the role of MSX1 in formation of uterus receptive during early pregnancy of goat, we investigated the expression and localization of MSX1 in theendometrium on day 5(P5), day 15(P15) and day 18(P18) of pregnancy.’ ……
Please add scale bar in all pictures of immunofluorescence in Fig. 2, 3, 4.
6. References, format of most references should be revised. For example, References 3, 4, 6, 7, 9-16, 18-22, 24, 25-41, 43-48, 50.
Author Response
We would like to thank the reviewer for their accurate and detailed revision of our manuscript. Reviewer offered helpful and constructive suggestions and as a result, we feel that the manuscript has been significantly improved. We have seriously thought about the suggestions and provided our responses in a point-by-point manner.

Reviewer 3 Report
The paper entitled “MSX1 regulates goat endometrial function by altering plasma membrane transformation of endometrial epithelium cells during early pregnancy” describes the complex role of MSX1 in the changes of the goat endometrial luminal epithelial cells polarity and sensitivity to the embryo during the maternal recognition of pregnancy and implantation. I found it interesting and comprehensive. However, it seems the Authors did not prepare the manuscript properly. Especially, the ”Material and methods” chapters lacks many required information. Although I found the paper valuable, it may not be accepted at the present form. Below I have listed all my doubts and comments:
1) The English grammar and style in the reviewed paper is of a very bad quality and should be corrected extensively.
2) The “Introduction” section provide sufficient reasoning for the presented study, however, I would suggest to describe the hypothesis and aim of the study more carefully. There is also no information on other treatment like TG and 4-PBA on knockdown of MSX1.
3) The manuscript lacks for spaces in many places i.e. before each citation and almost each semicolon. It seems that the Authors did not read and corrected the paper properly.
4) Please provide the number of the Ethics Committee approvement for this study.
5) There is not enough information on animals used in this study (like feeding, keeping conditions, pregnancy confirmation, way of insemination – natural/artificial, were they in their first or another pregnancy, how old were the animals, etc.)
6) The authors did nor express in any part of the manuscript why they chose these specific days of pregnancy. Please, remember that not every reader knows goat’s pregnancy physiology.
7) Please provide hosts, catalogue numbers, manufacturers and dilutions of all antibodies (primary and secondary) used in the study.
8) Section 2.2: How did the Authors stain the cell nuclei, there is no information in the text. Was it DAPI?
9) How were the negative controls for F-IHC performed? Please, provide pictures for negative controls to the figures.
10) Lines 112-114: What about the incubations with the substrate? Was it DAB? What was the enzyme conjugated to the secondary antibody?
11) What was the basis for P4 and E2 doses?
12) Section 2.3: How were the cells obtain? Did the Authors isolate them by themselves? Provide the procedure of isolation. When was actually IFNτ added to the culture? After 12h or at the same time as P4 and E2? Was it added alone or only in mixture with steroids? To be honest, the time line for the in vitro culture treatment addition is messy and without any chronology.
13) Section 2.4: ”What do you mean by “100-mesh”? 100 µm of diameter? What was the culture atmosphere?
14) Section 2.5: it seems that the total RNA was isolated not only from the EECs but from a mixture of EECs with GTCs. The whole method for transcript level studies should be described according to the MIQE guidelines. There are no basic information that would make the study reproducible: chemical and thermal conditions, concentrations of RNA for RT reaction and cDNA for qPCR, concentrations of primers, etc. Another question is that if the Authors design the primers for qPCR themselves? If so, they have to provide the results of primer validation study (i.e. reaction efficiency, slope of the standard curve, etc.). At this form it is inacceptable. What about the negative controls?
15) Section 2.6: What was the amount of total protein in each sample used for each experiment? What were the dilution of antibodies? What was the manufacturer of the HRP-substrate?
16) Section 2.7: line 173: what samples (cells, tissues)? What about the negative controls? How were they prepared and why the Authors did not show them? Please, add the negative controls pictures to the figures.
17) The Authors should explain each shortcut before used (i.e. shMSX1, shN, etc.)
18) Section 2.10: What was the reason for using two different post-hoc test for the same analyses?
19) Fig. 1: What was the reason for using “A” if there is only one panel. I guess that the following pictures are for biological replicates obtained from different animals on the same days of pregnancy. However, it should be marked in the figure, in the fig. heading, and in the text (add the information on the total number of animals used in the study. Pictures in the fourth line are of bad quality.
20) Lines 213-216: the heading is no sufficient. What is exactly on the following pictures, are these biological repetitions. It has to be explained in the text and marked in the figure.
21) Line 218: Which “hormone”? Please, specify. What do you mean by “and or”?
22) Lines 230-233: The data in the figure heading are not consistent with the “Methods” section. There are no information on 6h-incubation with IFNτ.
23) How did the Authors measure the protein expression of SPP1? F-IHC is a qualitative method not quantitative! In the “Methods” section you did not mention any details. To be honest, in Fig. 3 I see no difference in SPP1 expression. How many replicates did you preformed?
24) Fig. 3: The panels are to small ang graphs are of low quality. Letters are to small. The figure is hardly readable. There is no panel “I”.
25) Lines 273-274: Reduced? It seems that the expression is higher in E2+P4+ IFNτ samples.
26) Line 284 and 297: In fig. 4 and 5 there are no “ns” – why are they indicated in the headings?
27) Line 290: in fig. 4B there are no results on the knockdowns. I guess it should be 5B.
28) Fig. S1: Pictires in panel “C” are of very poor quality. The heading is not consistent with the panels (“A” is protein and “B” is mRNA).
29) Minor comments:
a) Line 44: “…endoplasmatic reticulum (ER) stress-mediated…”
b) Line 45: “…protein response (UPR) pathway.”
c) “in vivo” and ’in vitro” should be written italic
d) Line 93: “…was involved…”
e) Line 95: “…early pregnancy…”
f) Line 100: “The uterine tissues……from pregnant goats…”
g) Line 107: “…streptavidin-peroxidase…”
h) Line 149: “PrimerScript”
i) Line 315: “…in regulation of endometrial…”
j) Line 346: “… may be involved…”
k) Lines 357-369: this sentence has to rewritten as it is of a very bad style and hard to understand.
l) Line 384: explain “JAR”
m) Lines 403-404: “…may regulate the…”
n) Names of genes should be written italic.
o) In many sites the Authors use “quantitative” without “PCR”, it should be “quantitative PCR” or “qPCR”
Author Response
We would like to thank the reviewer for their accurate and detailed revision of our manuscript. Reviewer offered helpful and constructive suggestions and as a result, we feel that the manuscript has been significantly improved. We have seriously thought about the suggestions and provided our responses in a point-by-point manner.
Comments
The paper entitled “MSX1 regulates goat endometrial function by altering plasma membrane transformation of endometrial epithelium cells during early pregnancy” describes the complex role of MSX1 in the changes of the goat endometrial luminal epithelial cells polarity and sensitivity to the embryo during the maternal recognition of pregnancy and implantation. I found it interesting and comprehensive. However, it seems the Authors did not prepare the manuscript properly. Especially, the ”Material and methods” chapters lacks many required information. Although I found the paper valuable, it may not be accepted at the present form. Below I have listed all my doubts and comments:
Response: Thanks for reviewer’s positive comments and constructive suggestions of the work, we have seriously thought about the suggestions and provided our responses in a point-by-point manner, more required information and revisions were showed in the manuscript, especially in the “Material and methods”.
Point 1: The English grammar and style in the reviewed paper is of a very bad quality and should be corrected extensively.
Response 1: Thanks for reviewer’s kindly reminding and advice. We regret there were problems with the English. The manuscript has been polished before submission(the prove has been uploaded), but the manuscript may didn’t been carefully revised, so we have revised it seriously again as suggested, and carefully checked it.
Point 2: The “Introduction” section provide sufficient reasoning for the presented study, however, I would suggest to describe the hypothesis and aim of the study more carefully. There is also no information on other treatment like TG and 4-PBA on knockdown of MSX1.
Response 2: Thanks for reviewer’s constructive suggestions of the work.
- In the last paragraph, ‘However, the localization and expression of MSX1 has not been reported in uterus of goat, especially in P5, P15, P18. Moreover, whether MSX1 regulates the dynamic changes of gEECs remains poorly understood. Additionally, whether MSX1 can be induced by E2, P4 and IFNτ to participate in dynamic changes of gEECs remain unclear.’ was added for the hypothesis and aim.
- Thapsigargin (TG) and 4 phenyl butyric acid (4-PBA) are used as the activator and inhibitor of ER stress. In the present study, the function of MSX1 was explored in gEECs when the pathway of ER stress was affected, the relative information was showed in section 2.3 and the discussion of the manuscript.
Point 3: The manuscript lacks for spaces in many places i.e. before each citation and almost each semicolon. It seems that the Authors did not read and corrected the paper properly.
Response 3: Thanks for reviewer’s kindly reminding and advice, the space was added before each citation and almost each semicolon, which was revised in the whole manuscript. For example, ‘goat endometrial epithelial cells(gEECs)’ has revised into ‘goat endometrial epithelial cells (gEECs)’, ‘The receptive endometrium and filamentous conceptus are essential for implantation[1, 2].’ has revised into ‘The receptive endometrium and filamentous conceptus are essential for implantation [1, 2].’.
Point 4: Please provide the number of the Ethics Committee approvement for this study..
Response 4: Thanks for reviewer’s kindly reminding and advice. We have added the number (Approval No. 2019100903) of the Ethics Committee approvement of the study.
Point 5: There is not enough information on animals used in this study (like feeding, keeping conditions, pregnancy confirmation, way of insemination – natural/artificial, were they in their first or another pregnancy, how old were the animals, etc.).
Response 5: Thanks for reviewer’s kindly reminding and advice. We have refined the information of animals. Multiparous Guanzhong dairy goats (n=9, aged 2-3 years, average weight=59.28±1.93kg,) were reared on free feeding in the experimental animal center of Northwest A & F University, Yangling, China. The goats exhibiting at least two estrous cycles of normal duration were used in this study. At estrus, female goats were mated with fertile males to induce natural pregnancy, which was recorded as day 0 of pregnancy. Pregnancy was confirmed on day 5 by recovering blastocysts from the uterus. Pregnancy at day 15 and day 18 was respectively identified by observing the elongated tubular conceptus and fibrous conceptus in the uterus.
Point 6: The authors did nor express in any part of the manuscript why they chose these specific days of pregnancy. Please, remember that not every reader knows goat’s pregnancy physiology.
Response 6: Thanks for reviewer’s kindly reminding and advice. We have added the reasons for choosing these specific days of pregnancy in introduction and method sections of manuscript.
Introduction: During embryo implantation, the endometrium undergoes differentiation from pre-receptivity to receptivity in ruminant [1]. The development of endometrial receptivity is known as the “window of implantation” from the attachment of conceptus to the completion of adhesion [2,3]. On day 5 of pregnancy in goats, the endometrium was in a pre-receptivity stage. The endometrium was in a receptivity stage on day 15 [4,5]. The conceptus adheres firmly to the receptivity endometrium on day 18 of pregnancy in goats [6].
Method: The uterus samples were collected from pregnancy goats on day 5 (n=3, pre-receptivity endometrium), day 15 (n=3, receptive endometrium) and day 18(n=3, conceptus adhesion period) after the goats were subjected to midventral laparotomy and hysterectomy.
Point 7: Please provide hosts, catalogue numbers, manufacturers and dilutions of all antibodies (primary and secondary) used in the study.
Response 7: Thanks for reviewer’s kindly reminding and advice. We have revised the information of all antibodies in the manuscript.
(1) In sections 2.2
Primary antibody anti-MSX1 antibody (Human, ab168745, Abcam, USA, diluted 1:100)
Secondary antibody biotin-labeled goat anti-mouse IgG (Maixin-Biotech, Fuzhou, China)
(2) In section 2.6
Primary antibody anti-MSX1 antibody (Human, ab168745, Abcam, USA, diluted 1:1000)
Primary antibody anti-CDH2 antibody (Mouse, ab76057, Abcam, USA, diluted 1:1000)
Primary antibody anti-EIF2S1 (phosphorS51) antibody (Human, ab32157, Abcam, USA, diluted 1:1000)
Primary antibody anti-EIF2S1 antibody (Human, ab26197, Abcam, USA, diluted 1:1000)
Primary antibody anti-IRE1 (phosphorS724) antibody (Human, ab124945, Abcam, USA, diluted 1:1000)
Primary antibody anti-XBP1 antibody (Human, ab37152, Abcam, USA, diluted 1:1000)
Primary antibody anti-ATF6 antibody (Human, ab83504, Abcam, USA, diluted 1:1000)
Primary antibody anti-β-actin antibody (Mouse, Sanjian Biotech, Wuhan, China, diluted 1:5000)
Primary antibody anti-GRP78 antibody (Mouse, 3183, CST, USA, diluted 1:1000)
Secondary antibody HRP-labeled goat anti-rabbit or goat anti-mouse immunoglobulin (Sanjian Biotech, Wuhan, China, diluted 1:5000)
(3) In section 2.7
Primary antibody anti-MSX1 antibody (Human, ab168745, Abcam, USA, diluted 1:100)
Primary antibody anti-CDH2 antibody (Mouse, ab76057, Abcam, USA, diluted 1:100)
Primary antibody anti-SPP1 antibody (Human, WL02378, Wanleibio Co., Ltd, diluted 1:150)
Secondary antibody Alexa-labeled donkey anti-mouse IgG (mouse, A16016, Invitrogen, Waltham, MA, USA diluted 1:500)
Secondary antibody Alexa-labeled donkey anti-rabbit IgG (rabbit, A16028, Invitrogen, Waltham, MA, USA diluted 1:500)
Point 8: Section 2.2: How did the Authors stain the cell nuclei, there is no information in the text. Was it DAPI?.
Response 8: Thanks for reviewer’s kindly reminding and advice. The cell nuclei were stained with hematoxylin for 25 s, which have refined the information of cell nuclei staining in Section 2.2.
Point 9: How were the negative controls for F-IHC performed? Please, provide pictures for negative controls to the figures.
Response 9: Thanks for reviewer’s kindly reminding and advice. In the immunofluorescent staining experiments, the negative control refers to incubating only with the secondary antibody but not with the primary antibody, which was added in the supplement figure S3.
Point 10: Lines 112-114: What about the incubations with the substrate? Was it DAB? What was the enzyme conjugated to the secondary antibody?
Response 10: Thanks for reviewer’s kindly reminding and advice. We have refined the information on immunohistochemistry. After incubated with anti-MSX1 antibody (1:100) at 4℃ overnight, the sections incubated with biotin-labeled goat anti-mouse IgG at 25℃ for 2 h. After washing PBS, the sections were incubated with streptavidin-biotin peroxidase for 40 min at 25℃. Thereafter, the sections were visualized with diaminobenzidine (DAB).
Point 11: What was the basis for P4 and E2 doses?
Response 11: Thanks for reviewer’s raising this question. The basis for P4 and E2 doses are cited form previous studies of our laboratory [7,8], which was use to mick the hormonal environment of early pregnancy in vitro.
Point 12: Section 2.3: How were the cells obtain? Did the Authors isolate them by themselves? Provide the procedure of isolation. When was actually IFNτ added to the culture? After 12h or at the same time as P4 and E2? Was it added alone or only in mixture with steroids? To be honest, the time line for the in vitro culture treatment addition is messy and without any chronology.
Response 12: Thanks for reviewer’s kindly reminding and advice.
(1) The gEECs was obtained by transfection human telomerase reverse transcriptase with the primary endometrial epithelial cells of goat. The gGTCs was obtained by transfection human telomerase reverse transcriptase with the primary trophoblast cells of goat.
(2) The primary endometrial epithelial cells of goats were preserved in our laboratory. The methods of isolation was cited from the reference [9]. The uteri of dairy goat (8-month-old) were collected. The endometrium was prepared into small tissue blocks and digested by Collagenase â… for 4 h at 37℃. The postdigestive cells suspension was filtered by 200 cell sieve and the filtrate was centrifuged for 5 min, 500 rpm. After the low-speed centrifugalization, primary EECs were pelleted to the bottom. They were further purified by naturalsettling-low centrifugalization for 3-5 times.
The primary trophoblast cells of goats preserved in our laboratory. The methods of isolation was cited from the reference [10]. Pregnant dairy goat uteri (45–60 days of pregnancy) were collected. The fetal cotyledons were manually separated from the maternal caruncles and rinsed thoroughly in phosphate buffered saline (PBS), then the fetal cotyledons were minced into 1 mm3 pieces and disaggregated with 2 mg/mL collagenase Type I at 37°C for 30 min. The supernatants were filtered through 200 mesh stainless steel screens to remove undigested tissue fragments. The filtrate was centrifuged and resuspended with serum-free DMEM/F12 medium. Dissociated cells were purified by isopycnic centrifugation on density gradients between 30 % and 45 % Percoll.
(3) After gEECs were treated with P4 and E2 for 12h, IFNτ was added to medium and continued to treat for 6h or 12h.
(4) IFNτ was added only in mixture with steroids.
(5) We feel sorry for the problems in section 2.3. We have revised carefully the manuscript in section 2.3. In order to clearly describe the in vitro culture treatment addition, we drew a time image shown in Figure S1. please check it.
Point 13: P Section 2.4: ”What do you mean by “100-mesh”? 100 µm of diameter? What was the culture atmosphere?
Response 13: Thanks for reviewer’s kindly reminding and advice. 100-mesh means that there are 100 holes per square inch. The mesh size of 100 mesh screen is 0.150 mm. The cell suspension of gGTCs was cultured on an orbital shaker rotating at 200 rpm at 37℃ in a humidified 5 % CO2 incubator.
Point 14: Section 2.5: it seems that the total RNA was isolated not only from the EECs but from a mixture of EECs with GTCs. The whole method for transcript level studies should be described according to the MIQE guidelines. There are no basic information that would make the study reproducible: chemical and thermal conditions, concentrations of RNA for RT reaction and cDNA for qPCR, concentrations of primers, etc. Another question is that if the Authors design the primers for qPCR themselves? If so, they have to provide the results of primer validation study (i.e. reaction efficiency, slope of the standard curve, etc.). At this form it is inacceptable. What about the negative controls?
Response 14: Thanks for reviewer’s kindly reminding and advice.
(1) In section 2.5, the total RNA was isolated only from the gEECs. The gGTCs were only used for adhesion experiment to calculate adhesion rates.
(2) We have added the information in section 2.5 of the manuscript. Total RNA was extracted from gEECs using Trizol reagent (TaKaRa, Tokyo,Japan) according to the manufacturer’s instructions. The RNA concentration and purity were measured based on the method previously described [11]. Briefly, The extracted RNA was dissolved in diethypyrocarbonate (DEPC)-treated water, and the RNA concentration and purity were estimated by reading the absorbance at 260 and 280 nm on a spectrophotometer (Eppendorf, Inc., Hamburg, Germany). The absorption ratios (260/280 nm) for all preparations were between 1.8 and 2.0. The cDNA was synthesized using a PrimeScript™ RT reagent Kit (TaKaRa Bio, Inc., Dalian, China) according to the manufacturer’s instructions. The final volume of each experimental reaction was 20 µl, which included 400 ng of total RNA. Real-time quantitative PCR was carried out using SYBR Green Master Mix (Vazyme Bio, Inc., China) in Bio-Rad CFX96 (CFX, Bio-Rad Laboratories, Inc., Hercules, CA, USA) according to the manufacturer’s protocol. Each PCR reaction (total volume of 20 μL) consisted of 2 μL of cDNA, 0.8 μL of 10 μM forward and reverse primer each, 10 μL of SYBR® Premix Ex Taq™ II, and 7.2 μL of RNase-free water. The cycling conditions included a denaturation step at 95 °C for 30 s, followed by 40 PCR cycles at 95 °C for 5 s and 60 °C for 20 s. A melting curve analysis was performed at the end of each PCR programme to exclude the formation of nonspecific products.
(3) Some primers (MSX1, ZO-1, α-PKC, Par3, CDH2, SCRIB, Lgl2) are designed by ourselves, and we have provided the amplification curve and melt curve, which were show in supplement materials; other genes(PTGS, PTGS2, PTGES, PGFS, ISG15, RSAD2, CXCL10, SPP1, GAPDH) were cited from the previous research, which was showed in supplement table 1 of the manuscript.
Point 15: Section 2.6: What was the amount of total protein in each sample used for each experiment? What were the dilution of antibodies? What was the manufacturer of the HRP-substrate?
Response 15: Thanks for reviewer’s kindly reminding and advice. We have added the information in section 2.6 of the manuscript.
- 20 μg total protein were used in each sample for each experiment.
- anti-MSX1 antibody (1:1000, ab168745, Abcam, USA); anti-CDH2 antibody (1:1000, ab76057, Abcam, USA); anti-EIF2S1 (phosphorS51) antibody (1:1000, ab32157, Abcam, USA); anti-EIF2S1 antibody (1:1000, ab26197, Abcam, USA); anti-IRE1 (phosphorS724) antibody (1:1000, ab124945, Abcam, USA); anti-XBP1 antibody (1:1000, ab37152, Abcam, USA); anti-ATF6 antibody (1:1000, ab83504, Abcam, USA); anti-GRP78 antibody (1:1000, 3183, CST, USA); anti-β-actin antibody (1:5000, Sanjian Biotech, Wuhan, China).
- Secondary antibody HRP-labeled goat anti-rabbit or goat anti-mouse immunoglobulin (1:5000, Sanjian Biotech, Wuhan, China).
Point 16: Section 2.7: line 173: what samples (cells, tissues)? What about the negative controls? How were they prepared and why the Authors did not show them? Please, add the negative controls pictures to the figures.
Response 16: Thanks for reviewer’s kindly reminding and advice. We have refined the information of immunofluorescent staining.
- All the samples of immunofluorescent staining are cell sections.
- In our immunofluorescent staining experiments, the negative control refers to incubating only with the Alexa-labeled secondary antibody but not with the primary antibody.
(3) For figure 2 and figure 4: Cell sections were seeded in 24-well plates. gEECs were plated into 24-well plates (1×104cells/well) and cultured in DMEM/F-12 medium containing 10 % FBS at 37 °C in a humidified 5 % CO2 incubator. When gEECs reached 50-60 %, the medium was replaced with fresh DMEM/F-12 supplemented 0.1 % bovine serum albumin for 24h. After the gEECs were treated with P4 and E2 for 12h, IFNτ was added into the medium for 6h. Thereafter, the cell sections were fixed with 4 % formaldehyde.
For figure 3: Cell sections were seeded in 24-well plates. gEECs were infected with lentiviral vectors encoding the MSX1 shRNA (shMSX1) or negative control short hairpin RNA (shN), then were plated into 24-well plates (1×104cells/well) and cultured in DMEM/F-12 medium containing 10 % FBS at 37 °C in a humidified 5 % CO2 incubator. When cells reached 50-60 %, the medium was replaced with fresh DMEM/F-12 supplemented 0.1 % bovine serum albumin for 24h. After the gEECs were treated with P4 and E2 for 12h, IFNτ was added to the medium for 6h. Thereafter, the cell sections were fixed with 4 % formaldehyde.
Point 17: The Authors should explain each shortcut before used (i.e. shMSX1, shN, etc.).
Response 17: Thanks for reviewer’s kindly reminding and advice. We have revised the shortcuts in the manuscript.
shMSX1: recombinant lentiviral vectors encoding the MSX1 shRNA (shMSX1)
shN: negative control short hairpin RNA (shN)
Point 18: Section 2.10: What was the reason for using two different post-hoc test for the same analyses?
Response 18: Thanks for reviewer’s kindly reminding and advice. The method was cited from the reference [7]. LSD method was used for the higher sensitivity in the pairwise comparison method than others(Sidak method, Bonferroni method, Scheffe method, Dunnett method) , and the turkey method was used for the same sample size in this research, so we considered the two different post-hoc test for the one-way analysis of the manuscript
LSD method: Least Significance Difference(LSD) method is one of the simplest comparison methods. It is a simple deformation of t test, and does not make any correction to the test level, but estimates a more robust standard error by fully considering all sample information of the population level in the calculation of standard error (note not standard deviation). The LSD method is the most sensitive post-hoc multiple comparison method because the significance level a of the single comparison remains the same.
Tukey method: Tukey's Honestly Significant Difference law. The application of this method requires the same amount of samples in each group. It also uses Studentized Range distribution to compare the means of each group, it controls that the maximum probability of "truth-rejecting" error in all comparisons does not exceed the set significance level a.
Point 19: Fig. 1: What was the reason for using “A” if there is only one panel. I guess that the following pictures are for biological replicates obtained from different animals on the same days of pregnancy. However, it should be marked in the figure, in the fig. heading, and in the text (add the information on the total number of animals used in the study. Pictures in the fourth line are of bad quality.
Response 19: Thanks for reviewer’s professional review work of the manuscript. We regret there were problems with heading and typography of the image in figure 1. According to your suggestions, we have made extensive corrections to our previous draft, the detailed corrections are listed below.
3.1 Immunolocalization of MSX1 in the endometrial tissue of goats
Representative images of MSX1 immunolocalization in goat endometrial tissue at pregnancy 5 (P5, n=3), 15(P15, n=3) and 18 (18, n=3) were shown in figure 1A-I. The immunohistochemical staining of MSX1 in endometrial tissue showed that MSX1 was mainly localized in LE(Fig. 1D-F), sGE(Fig. 1D-F) and GE(Fig. 1G-I), while a small amount of expression was detected in the stromal cells (S) (Fig. 1D-F). Compared with P5, the immunoreactivity of MSX1 was upregulated at P15 and P18.
Point 20: Lines 213-216: the heading is no sufficient. What is exactly on the following pictures, are these biological repetitions. It has to be explained in the text and marked in the figure.
Response 20: Thanks for reviewer’s kindly reminding and advice. According to the suggestion, we have made corrections to our previous draft, the detailed corrections are listed below.
Figure 1. Representative images of MSX1 immunolocalization in goat endometrial tissue at pregnancy 5, 15 and 18. (A,D,G) Representative images of MSX1 immunolocalization in goat endometrial tissue on P5., (B,E,H) Representative images of MSX1 immunolocalization in goat endometrial tissue on P15. (C,F,I) Representative images of MSX1 immunolocalization in goat endometrial tissue on P18. (G,K,L). Representative images of the negative control in goat endometrial tissue on P5, P15, P18. LE: luminal epithelial cells, sGE: superficial glandular epithelium, GE: glandular epithelial cells, S: stromal cells, m: myometrium; A-C, scale bar=200 μ1m; D-L, scale bar=50 μm.
Point 21: Line 218: Which “hormone”? Please, specify. What do you mean by “and or”?
Response 21: Thanks for reviewer’s kindly reminding and advice. Hormone refer to ‘E2 and P4, the ‘and or’ refer to ‘and/or’, we have revised in the manuscript.
Point 22: Lines 230-233: The data in the figure heading are not consistent with the “Methods” section. There are no information on 6h-incubation with IFNτ.
Response 22: Thanks for reviewer’s kindly reminding and advice. We have revised the manuscript in section 2.3, the detailed corrections are listed below.
Lines 122-125: The EECs were then treated with P4 (10-7 M; Sigma, St. Louis, MO, USA) and/or E2 (10-9 M, Sigma, St. Louis, MO, USA) for 12 h. IFNτ (20 ng/mL, Sangon Biotech Co., Ltd, Shanghai, China) was added to the medium and continued to treat for 6h or 12h.
Point 23: How did the Authors measure the protein expression of SPP1? F-IHC is a qualitative method not quantitative! In the “Methods” section you did not mention any details. To be honest, in Fig. 3 I see no difference in SPP1 expression. How many replicates did you preformed?
Response 23: Thanks for reviewer’s kindly reminding and advice. We have added the information of SPP1 in section 2.7 and Table S1 of manuscript. Meanwhile, the quantitative analysis of SPP1 fluorescence intensity was added in figure 3.
(1) We measured the protein expression of SPP1 through quantitative analysis of immunofluorescence, and it also has been used as a method to detect protein expression in some literature [12].
(2) Primary antibody anti-SPP1 antibody (1:150, WL02378, Wanleibio Co., Ltd,). This antibody has been reported for goats, and the protein level of SPP1 were detected by immunofluorescence method in these references [7,8,13]. However, the antibody is not available for western blot, so the protein levels were not detected by western blot .
(3) Representative images of three independent experiments are shown. The data of quantitative analysis are presented from three independent experiments.
Point 24: Fig. 3: The panels are to small ang graphs are of low quality. Letters are to small. The figure is hardly readable. There is no panel “I”.
Response 24: Thanks for reviewer’s kindly reminding and advice.
- We have adjusted the font size and pixels in figure 3, which have revisedin the manuscript.
- “I” has revised into “H” in figure 3.
Point 25: Lines 273-274: Reduced? It seems that the expression is higher in E2+P4+ IFNτ samples.
Response 25: Thanks for reviewer’s kindly reminding and advice, it’s indeed a mistake. In Figure 4C, the fluorescence intensity of cytoplasmic CDH2 was enhanced in E2 + P4 + IFNτ group compared with the control group, which have revised in the manuscript.
Point 26: Line 284 and 297: In fig. 4 and 5 there are no “ns” – why are they indicated in the headings?
Response 26: Thanks for reviewer’s kindly reminding and advice, it’s indeed a mistake. We have detected the ‘“ns” as p>0.05’ in fig.4 and 5.
Point 27: Line 290: in fig. 4B there are no results on the knockdowns. I guess it should be 5B.
Response 27: Thanks for reviewer’s kindly reminding and advice, it’s indeed a mistake. We have revised the sentence into ‘these results showed that the knockdown of MSX1 decreased the expression of CDH2 following E2, P4, and IFNτ treatment (Fig. 5A and 5B, p<0.01)’.
Point 28: Fig. S1: Pictires in panel “C” are of very poor quality. The heading is not consistent with the panels (“A” is protein and “B” is mRNA).
Response 28: Thanks for reviewer’s kindly reminding and advice. We have revised the section in figure S1, which showed as follows:
Figure S1. Interference efficiency of shMSX1 vectors. (A) The representative images and quantitative analysis of MSX1 protein level. (B) The quantitative analysis of MSX1 mRNA level. (C-D) The representative confocal microscopeimages and quantitative analysis of MSX1 expression in shN group and shMsx1group. “*” as P<0.05; “**” as P<0.01.
Point 29: Minor comments:
- a) Line 44: “…endoplasmatic reticulum (ER) stress-mediated…”
Response a: Thanks for reviewer’s kindly advice. We have revised “…endoplasmatic reticulum stress-mediated…” into “…endoplasmatic reticulum (ER) stress-mediated…”.
- b) Line 45: “…protein response (UPR) pathway.”
Response b: Thanks for reviewer’s kindly reminding and advice. We have revised “……protein response pathway.” into “…protein response (UPR) pathway.”.
- c) “in vivo” and ’in vitro” should be written italic
Response c: Thanks for reviewer’s kindly reminding and advice. We have revised “in vitro” into “in vitro” in the manuscript, mainly in section 2.4.
- d) Line 93: “…was involved…”
Response d: Thanks for reviewer’s kindly reminding and advice. We have revised the manuscript in line 93.
- e) Line 95: “…early pregnancy…”
Response e: Thanks for reviewer’s kindly reminding and advice. We have revised the manuscript in line 95.
- f) Line 100: “The uterine tissues……from pregnant goats…”
Response f: Thanks for reviewer’s kindly reminding and advice. We have revised “The uterus tissues……from pregnant goats…” into “The uterine tissues……from pregnant goats…” in the manuscript of section 2.1.
- g) Line 107: “…streptavidin-peroxidase…”
Response g: Thanks for reviewer’s kindly reminding and advice. We have revised “…streptavidin-perosidase…” into “…streptavidin-peroxidase…” in the manuscript of section 2.2.
- h) Line 149: “PrimerScript”
Response h: Thanks for reviewer’s kindly reminding and advice. We have revised “PrimerScripyt” into “PrimerScript” in the manuscript of section 2.5.
- i) Line 315: “…in regulation of endometrial…”
Response i: Thanks for reviewer’s kindly reminding and advice. We have revised “…in regulate endometrial…” into “…in regulation of endometrial…” in the manuscript of section 3.7.
- j) Line 346: “… may be involved…”
Response j: Thanks for reviewer’s kindly reminding and advice. We have revised “… may involve…” into “… may be involved…” in the manuscript of discussion.
- k) Lines 357-369: this sentence has to rewritten as it is of a very bad style and hard to understand.
Response k: Thanks for reviewer’s kindly reminding and advice. We have revised the sentences in lines 357-369 into“In ruminants, some genes that promote conceptus elongation, such as ISG15, CXCL10, and RSAD2, which are considered markers for endometrial receptivity formation” in the manuscript of discussion.
- l) Line 384: explain “JAR”
Response l: Thanks for reviewer’s kindly reminding and advice. The “JAR” refers to “a human choriocarcinoma cell line” [14].
- m) Lines 403-404: “…may regulate the…”
Response m: Thanks for reviewer’s kindly reminding and advice. We have revised “…may be regulate the…” into “…may regulate the…” in the manuscript of discussion
- n) Names of genes should be written italic.
Response n: Thanks for reviewer’s kindly reminding and advice. The names of genes have revised into “italic” in the whole manuscript.
- o) In many sites the Authors use “quantitative” without “PCR”, it should be “quantitative PCR” or “qPCR”.
Response o: Thanks for reviewer’s kindly reminding and advice. We have revised “quantitative” into “qPCR” in the whole manuscript.
Reference
1 Spencer TE, Johnson GA, Bazer FW, Burghardt RC, Palmarini M: Pregnancy recognition and conceptus implantation in domestic ruminants: roles of progesterone, interferons and endogenous retroviruses. Reproduction, fertility, and development 2007;19:65-78.
2 Lessey BA, Young SL: What exactly is endometrial receptivity? Fertility and sterility 2019;111:611-617.
3 Spencer TE, Johnson GA, Bazer FW, Burghardt RC: Implantation mechanisms: insights from the sheep. Reproduction (Cambridge, England) 2004;128:657-668.
4 Igwebuike UM: A review of uterine structural modifications that influence conceptus implantation and development in sheep and goats. Anim Reprod Sci 2009;112:1-7.
5 Zhang L, An XP, Liu XR, Fu MZ, Han P, Peng JY, Hou JX, Zhou ZQ, Cao BY, Song YX: Characterization of the Transcriptional Complexity of the Receptive and Pre-receptive Endometria of Dairy Goats. Scientific reports 2015;5:14244.
6 Liu H, Wang C, Li Z, Shang C, Zhang X, Zhang R, Wang A, Jin Y, Lin P: Transcriptomic Analysis of STAT1/3 in the Goat Endometrium During Embryo Implantation. Front Vet Sci 2021;8:757759.
7 Yang D, Jiang T, Liu J, Hong J, Lin P, Chen H, Zhou D, Tang K, Wang A, Jin Y: Hormone regulates endometrial function via cooperation of endoplasmic reticulum stress and mTOR-autophagy. 2018;233:6644-6659.
8 Yang D, Jiang T, Liu J, Zhang B, Lin P, Chen H, Zhou D, Tang K, Wang A, Jin Y: CREB3 regulatory factor -mTOR-autophagy regulates goat endometrial function during early pregnancy. Biology of reproduction 2018;98:713-721.
9 YanYan Z, AiHua W, QingXia W, HongXia S, YaPing J: Establishment and characteristics of immortal goat endometrial epithelial cells and stromal cells with hTERT. Journal of animal and veterinary advances 2010;9:2738-2747.
10 Dong F, Huang Y, Li W, Zhao X, Zhang W, Du Q, Zhang H, Song X, Tong D: The isolation and characterization of a telomerase immortalized goat trophoblast cell line. Placenta 2013;34:1243-1250.
11 Lin P, Lan X, Chen F, Yang Y, Jin Y, Wang A: Reference gene selection for real-time quantitative PCR analysis of the mouse uterus in the peri-implantation period. PloS one 2013;8:e62462.
12 Coffman VC, Wu JQ: Counting protein molecules using quantitative fluorescence microscopy. Trends Biochem Sci 2012;37:499-506.
13 Yang D, Zhang B, Wang Z, Zhang L, Chen H: COPS5 negatively regulates goat endometrial function via the ERN1 and mTOR-autophagy pathways during early pregnancy. 2019;234:18666-18678.
14 Uchida H, Maruyama T, Nishikawa-Uchida S, Oda H, Miyazaki K, Yamasaki A, Yoshimura Y: Studies using an in vitro model show evidence of involvement of epithelial-mesenchymal transition of human endometrial epithelial cells in human embryo implantation. The Journal of biological chemistry 2012;287:4441-4450.

Round 2
Reviewer 1 Report
1. Figure 1: The WB and mRNA detection of MSX1 should be added. The total tissue protein or RNA is ok, not required for the extraction of the specific cells.
2. Figure 2B, Figure 4B, Figure 5A, and Figure 6A should show the gray values of all gels.
Author Response
We would like to thank the reviewer for their accurate and detailed revision of our manuscript. Reviewer offered helpful and constructive suggestions and as a result, we feel that the manuscript has been significantly improved. We have seriously thought about the suggestions and provided our responses in a point-by-point manner, are given as follows.
Point 1: Figure 1: The WB and mRNA detection of MSX1 should be added. The total tissue protein or RNA is ok, not required for the extraction of the specific cells.
Response 1: Thanks for reviewer’s kindly reminding and detailed advice again. In the present study, the WB and mRNA detection of MSX1 expression at 5, 15 and 18 days of pregnancy goat were added in figure 1B and 1C, results shown that the expression of MSX1 was significantly increased at 15 and 18 days of pregnancy goat compared with 5 days of pregnancy goats, please check it in figure 1B and 1C of manuscript.
Point 2: Figure 2B, Figure 4B, Figure 5A, and Figure 6A should show the gray values of all gels.
Response 2: Thanks for reviewer’s advice, and we have added gray values of all gels, which mainly revised in Figure 1C, Figure 2B, Figure 4B, Figure 5A, Figure 6A and Figure 7A-D, please check it in the manuscript.
Reviewer 3 Report
The Authors addressed all my doubts and questions. I would recommend the paper for publication in its present form.
Author Response
Comments
The Authors addressed all my doubts and questions. I would recommend the paper for publication in its present form.
Response: Thanks for reviewer’s positive comments of the work and recommendation for publication, thanks again for the previous constructive comments, the accurate and detailed revision of the manuscript, which has improved the study significantly.
Round 3
Reviewer 1 Report
-
There are still some grammar problems, please proofread carefully.